# Training with More Confidence: Mitigating Injected and Natural Backdoors During Training

**Zhenting Wang**
Rutgers University
zhenting.wang@rutgers.edu

**Hailun Ding**
Rutgers University
hailun.ding@rutgers.edu

**Juan Zhai**
Rutgers University
juan.zhai@rutgers.edu

**Shiqing Ma**
Rutgers University
sm2283@rutgers.edu

## Abstract

The backdoor or Trojan attack is a severe threat to deep neural networks (DNNs). Researchers find that DNNs trained on benign data and settings can also learn backdoor behaviors, which is known as the natural backdoor. Existing works on anti-backdoor learning are based on weak observations that the backdoor and benign behaviors can differentiate during training. An adaptive attack with slow poisoning can bypass such defenses. Moreover, these methods cannot defend natural backdoors. We found the fundamental differences between backdoor-related neurons and benign neurons: backdoor-related neurons form a hyperplane as the classification surface across input domains of all affected labels. By further analyzing the training process and model architectures, we found that piece-wise linear functions cause this hyperplane surface. In this paper, we design a novel training method that forces the training to avoid generating such hyperplanes and thus remove the injected backdoors. Our extensive experiments on five datasets against five state-of-the-art attacks and also benign training show that our method can outperform existing state-of-the-art defenses. On average, the ASR (attack success rate) of the models trained with NONE is 54.83 times lower than undefended models under standard poisoning backdoor attack and 1.75 times lower under the natural backdoor attack. Our code is available at https://github.com/RU-System-Software-and-Security/NONE.

## 1 Introduction

Deep Neural Networks (DNNs) are vulnerable to Trojans[1]. A Trojaned model makes normal predictions on benign inputs and outputs the target label when the input contains a specific pattern (i.e., Trojan trigger) such as a yellow pad. To inject a Trojan [1–7], the adversary can poison the training dataset by adding poisoning samples (or Trojan samples): inputs stamped with the Trojan trigger and labeled as the target label. This is a typical data poisoning attack, and the model can learn the trigger as a strong feature for the target label. Recently, researchers found the existence of *natural Trojans*. Namely, a model trained on benign datasets with normal settings (e.g., hyperparameters, optimizers) can also learn Trojans, when there exists a strong input pattern in the training dataset that corresponds to one label [8]. In such natural Trojan scenarios, the input pattern serves as a Trojan trigger, and its corresponding label is the target label. By reverse-engineering the trigger, the adversary can leverage it for attacks. As such, both injected and natural Trojans are severe threats.

---

[1]Trojan attack is also known as the backdoor attack in the existing literature.

36th Conference on Neural Information Processing Systems (NeurIPS 2022).

There are no existing works for learning a robust DNN against both injected and natural Trojans. Existing works focus on training benign classifiers when the dataset is poisoned. For example, ABL [9] observes that the model will learn backdoor behavior faster than benign behavior, and proposes a training algorithm to suppress learning the trigger pattern. DP-SGD is an optimization method that leverages the differential privacy (DP) method and combines it with stochastic gradient descent (SGD) to learn a robust classifier using poisoned datasets. These methods fail to defend against the natural Trojan.

In this paper, we propose a robust training algorithm that can mitigate both injected and natural Trojans. For a given Trojan in the form of $\tilde{x} = T(x, m, t) = (1 - m) \odot x + m \odot t$, where $\tilde{x}$ and $x$ are respectively the poisoning and benign samples, $\tilde{x}$ is the poisoning sample generation method using the trigger $(m, t)$ with trigger mask matrix $m$ and trigger value matrix $t$, we theoretically prove that there exists one and only one hyperplane in the input space that corresponds to all poisoning samples. Thus, the trained classifier is mapping this hyperplane to a target label when performing Trojan attacks. Fig. 1 intuitively illustrates our idea using a simple exam-

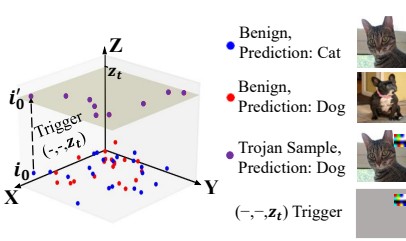

**Fig. 1:** Decision Region of A Trojan Model.

ple. To simplify the problem, each input has three dimensions $(d_x, d_y, d_z)$. We use red and blue dots to denote inputs belonging to different labels, and the trigger is denoted as $t = (-, -, z_t)$. Adding the trigger into an input $i_0 = (x_0, y_0, z_0)$ to get the corresponding Trojaned input $i'_0 = (x_0, y_0, z_t)$ is equivalent to moving this input to the $z = z_t$ plane in the input domain. As shown in Fig. 1, the input $i_0$ moves along the dashed line and ends up in the wheat colored plane, $z = z_t$. Notice that $i_0$ can be any input, and stamping the trigger will move them to the plane $z = z_t$. In other words, the plane $z = z_t$ contains all Trojan samples. Likewise, if there exists a plane $z = z_t$ that is a decision region, its corresponding input pattern $(-, -, z_t)$ is a Trojan trigger. Stamping such a trigger to any input is essentially projecting the input to this plane, and because all the inputs in this plane have the same label, it is equivalent to performing a Trojan attack. Extending this to the high dimensional space, the Trojan region will be a hyperplane, and the decision boundaries it shares with other regions will be linear. In summary, we can say that a Trojan in a DNN always pairs with a hyperplane as its Trojan region. Considering that modern DNNs are non-linear and non-convex, this rarely happens for benign models. By further analysis, we found that this is related to the use of activation functions. Modern DNNs tend to use piece-wise linear functions as their activation functions. Even though the function itself is linear, its sub-functions are linear. For example, one of the most popular activation function, ReLU (i.e, $y = \max(0, x)$), consists of two linear functions (i.e., $y = 0, x \leq 0$ and $y = x, x > 0$). When a model's weights and biases are trained to specific regions, the neuron values before activation functions will fall into the input domain of only one sub-function (e.g., $x > 0$). As a result, the output and input will form a linear relationship. Consequently, the model can generate a hyperplane decision region in the input domain. In other words, we have a hyperplane, denoted as $< x_0, x_1, x_2 > = < a_0, a_1, a_2 >$, as a decision region, and for a given input $i$, if we replace its values in dimensions $x_0, x_1$ and $x_2$ to $a_0, a_1$ and $a_2$, respectively, we can turn its output label to a desired one. A model training process includes randomness (e.g., random initiation values, optimization), which we cannot avoid. Many possible decision regions will give us the same or similar training/validation accuracy. Some training will learn a linear decision region while others will not. This explains the cause of DNN Trojans and answers the former question, which moves forward our understanding of DNN Trojans one more step.

Based on this analysis, we develop a revised training method, NONE (**NON**-Lin**E**arity) that identifies linear decision regions, filters out inputs that are potentially poisoned, and resets affected neurons to enforce non-linear decision regions during training. We evaluated our prototype built with Python and PyTorch on MNIST, GTSRB, CIFAR, ImageNet, and the TrojAI dataset. Compared with SOTA methods, NONE is more effective and efficient in mitigating different Trojan attacks (i.e., single-target attack, label-specific attack, label-consistent attack, natural Trojan attack, and the hidden trigger attack). For example, on average, the ASR (attack success rate) of the models trained with NONE is 54.83 times lower than undefended models under standard poisoning Trojan attack and 1.75 times lower under the natural Trojan attack.

Our contributions in this paper can be summered as follows. We analyze the cause of (injected and natural) DNN Trojans and conclude that linearity in DNN decision regions is the main reason. Further, we analyzed the source of linear DNN decision regions and explained why and when it happens using commonly used layers and activation functions. Then, we propose a novel and general revised training framework, NONE that detects and fixes injected and natural Trojans in DNN training. To the best of our knowledge, we are the first to defend natural Trojans. We evaluate NONE on five different datasets and five different Trojans and compare it with other SOTA techniques. Results show that NONE significantly outperforms these prior works in practice.

## 2  Related Work

**Trojans in DNN.** Prior work [1, 10–14, 2, 3, 15–18] demonstrates that the attackers can inject Trojans into the victim models by poisoning the training dataset. Later, researchers found that a DNN trained on the benign dataset with standard training procedure can also learn such Trojans, which is known as natural Trojans. By using reverse engineering methods designed for poisoned models, researchers were able to find natural triggers in pretrained models. For example, in ABS [8], authors show that a Network in Network (NiN) [19] model trained on a benign CIFAR-10 dataset has natural Trojans. Other works [14, 20] also documented similar findings in other models.

**Trojan Defense.** One way to defend against Trojan attacks is to filter out poisoning data before training [21, 22]. Poison suppression defenses [23, 24] restrain the malicious effects of poisoning samples in the training phase. Du et al. [23] and Hong et al. [24] apply DP-SGD [25] to depress the malicious gradients brought from poisoned samples. However, existing anti-backdoor learning methods are based on weak observations and they do not find the root cause of Trojans. They can be bypassed by adaptive attacks. In addition, they can not defend natural Trojans. For example, the SOTA anti-backdoor learning method ABL [9] is based on the observation that learning of backdoor behaviors and benign behaviors are distinctive. In detail, the training loss on backdoor examples drops much faster than that on benign examples in the first few epochs. It can be bypassed by slow poisoning attacks. More specifically, when the backdoor attack is label-specific (i.e., the samples with different original labels have different target labels) with low poisoning ratios, the model will learn backdoor behavior slower. We run ABL on the label-specific BadNets attack [1] with ResNet18 model and GTSRB dataset. The results show that the attack success rate is 93.93%, meaning that ABL can not defend against such an attack. This is not surprising because the label-specific attack is more complex, and it is based on the benign behavior of the models (i.e., the model needs to classify the original label of the backdoor examples first, then convert it to corresponding target labels). Another line of work tries to detect if a model has Trojan or not before its deployment. Model diagnosis based defenses [26–35] determine if a given model has a Trojan or not by inspecting the model behavior. Model reconstruction based defenses try to eliminate injected Trojans in infected models [36–40], which requires retraining the model with a set of benign data. Another approach is to defend Trojan attacks at runtime. Testing-based defenses [41–43] judge if the given input contains trigger patterns and reject the ones that are malicious.

## 3  Methodology

**Threat Model.** In this paper, we consider training time defense against Trojan attacks, which is also used by existing works [24, 22, 21]. The training dataset can contain poisoning samples (to inject intended Trojans) or benign (but leads to natural Trojans). As defenders, we control the training process but do not assume control over training datasets. Adopted from existing work [26, 41, 42, 27], Trojan sample generation can be formalized as Eq. 1, where $x$ and $\tilde{x}$ respectively are the benign input and Trojan input. $m$, $t$ respectively represent, mask of the trigger (i.e., whether a pixel is in the trigger region) and contents of the trigger. $\odot$ is the element-wise multiplication operation on two vectors, i.e., the Hadamard product.

$$\tilde{x} = T(x, m, t) = (1 - m) \odot x + m \odot t \tag{1}$$

### 3.1 Trojan Analysis

To facilitate our discussion, we first define *decision region* which includes all samples with the same predicted label. Formally, we define it as:

**Definition 3.1.** (Decision Region) For a deep neural network $\mathcal{M} : \mathcal{X} \mapsto \mathcal{Y}$ where $\mathcal{X}$ is the input domain $\mathbb{R}^m$ and $\mathcal{Y}$ is a set of labels $\{1 \dots k\}$, a decision region is an input space $\mathcal{R}^l \subseteq \mathcal{X}$, s.t., $\forall \boldsymbol{x} \in \mathcal{R}^l, \mathcal{M}(\boldsymbol{x}) = l$.

In most tasks, decision regions with the same label spread over the whole input space because natural inputs belonging to the same label are naturally distributed in this way. Similarly, we define *Trojan Decision Region*, or in short, we call it the *Trojan Region*, which is a subregion of the target label decision region:

**Definition 3.2.** (Trojan Region) For a Trojaned deep neural network $\mathcal{M} : \mathcal{X} \mapsto \mathcal{Y}$ with target label $l$, its Trojan regions are input spaces where $\mathcal{T} \subseteq \mathcal{R}^l$, s.t., all Trojan inputs $\tilde{\boldsymbol{x}} \in \mathcal{T}$, and all inputs in $\mathcal{T}$ are Trojan inputs.

Based on the definitions, we have the following theorem:

**Theorem 3.3.** *Given a model $\mathcal{M} : \mathcal{X} \mapsto \mathcal{Y}$ with the Trojan trigger $(\boldsymbol{m}, \boldsymbol{t})$, if the attack is complete (100% attack success rate) and precise (no other triggers), there exists one and only one hyperplane $\{\boldsymbol{Ax} - \boldsymbol{b} = 0\}$ Trojan region, where $i \in \{1 \dots m\}$, diagonal matrix $\boldsymbol{A}_{i,i} = \boldsymbol{m}_i, \boldsymbol{b} = \boldsymbol{At}$.*

The proof for Theorem 3.3 and empirical results are in Appendix (§ 8.1). The theorem shows that when a model learns a Trojan, it essentially learns a hyperplane as a decision region. Based on the definition of decision regions, we know that they are inverse functions of the model. Thus, the inverse function of the Trojan is a hyperplane. To understand **how** popular model architectures learn such hyperplanes in practice, we perform further analysis.

We start our discussion from typical Convolutional Neural Networks (CNNs). A convolutional layer with activation functions can be represented as $y_j = \sigma(\mathbf{W}_j^{\mathbf{T}} x_j + \mathbf{b}_j^{\mathbf{T}})$, where $x_j$ and $y_j$ are the inputs and outputs of layer $j$, $\mathbf{W}_j$ and $\mathbf{b}_j$ are trained weights and bias values, and $\sigma$ represents the activation function which is used to introduce non-linearity in this layer. Most commonly used activation functions, e.g., ReLU, are piece-wise linear. For example, ReLU is defined as $\sigma(x) = \max(0, x)$, which consists of two linear pieces separated at $x = 0$. As pointed out by Goodfellow et al. [44], even non-piece-wise linear functions are trained to semi-piece-wise linear. This helps resolve the gradient explosion/vanishing problem and makes training DNNs more feasible.

We make a key observation that *DNN Trojans will increase linearity of a convolutional layer with activation functions by introducing a large percentage of neurons activating on one piece of the activation function*. Specifically, we observe that when the Trojan behavior happens, the neuron values before activation functions fall into one input range of the activation function, which makes it linear. Recall that most well-trained activation functions are piece-wise linear, and if the inputs are in one input range, it regresses to a linear function. For example, layer $j$ using ReLU as its activation function will be a linear layer if $\mathbf{W}_j^{\mathbf{T}} x_j + \mathbf{b}_j^{\mathbf{T}} \geq 0$ for all $x_j$. As such, the reverse function of the Trojan can be a hyperplane or overlaps significantly with the hyperplane.

Fig. 2 shows the empirical comparison of a benign model and a Trojan model. In our experiment, we train a benign model $\mathcal{M}$ and a Trojan model $\mathcal{M}'$ using ResNet18 on CIFAR-10. While training $\mathcal{M}'$, we adopt the TrojanNet [45] training method which guarantees that only certain neurons will contain the Trojan, which we call Trojan neurons. The x-axis in Fig. 2 shows the value of $\mathbf{W}^{\mathbf{T}} x + \mathbf{b}^{\mathbf{T}}$, and y-axis shows the percentage of Trojan neurons whose activation value is

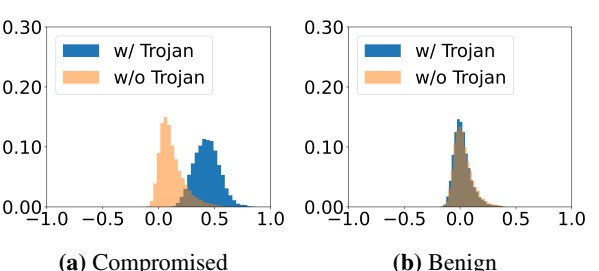

**(a)** Compromised      **(b)** Benign

**Fig. 2:** Comparison of Activation Values.

the corresponding value on x-axis. We use blue color to denote experiments when inputs have the trigger and orange to denote when inputs do not contain the trigger. As we can see, the model that

**Algorithm 1** Training

**Input:**    Training Data: $D$, Maximal epoch: $E$
**Output:**    Model: $M$

```
 1: function TRAINING(D)
 2:     while e ≤ E and (not terminate(M)) do
 3:         ▷ Train and gather activation values
 4:         M = train(D, e)
 5:         A = M.predict(D)
 6:         ▷ Identify compromised neurons
 7:         C = ∅
 8:         for neuron n in M do
 9:             if ℙ(Aₙ ≥ 0) ≥ θ  then
10:                 C = C ∪ {n}
11:         ▷ Identify biased or poisoning samples
12:         for neuron n ∈ C do
13:             Bₙ, Oₙ = separate(Aₙ)
14:             μ, σ = norm(Bₙ)
15:             for activation value of input i : iₙ ∈ Oₙ do
16:                 if |（iₙ−μ)/σ| ≥ λ then
17:                     D = D − i
18:         ▷ Resetting compromised neurons
19:         for neuron c ∈ C do
20:             c = M.init(c)
```

contains the Trojan will activate the $x > 0$ region when the input contains the trigger. By contrary, all the other three cases do not have such a phenomenon. We also conduct similar experiments on other different models and different types of layers including non-piece-wise linear layers, and all empirical results confirm our observation here. Details are in Appendix (§ 8.2).

## 4    Training Algorithm

To solve the Trojan problem caused by high linearity neurons, we propose Algorithm 1 to enforce non-linearity in individual layers by resetting potentially linear neurons and removing data samples that cause such linearity. This is a revised training process, which is an iterative process that trains the model until the maximal epochs (line 2). It starts by training the model using the standard backward propagation method (line 4). We also gather all activation values of individual neurons $A$ for all training samples in the training dataset $D$. Then, the process contains three steps: identifying compromised neurons (lines 6 to 10), identifying biased or poisoning samples (lines 11 to 17), and lastly, resetting the neurons for retraining (lines 18 to 20).

The first step is to identify compromised neurons, namely the neurons that carry Trojans. Based on our discussion on § 3.1, we do this by checking the activation values of neuron $n$, denoted as $A_n$, to see if its function is highly linear using the condition $\mathbb{P}(A_n \geq 0) \geq \theta$. If so, we make the neuron $n$ as potentially compromised and add it to the candidate set $C$. The second step is to identify highly biased samples or poisoning samples. The overall design is a statistical testing process: we first find a reference distribution of a particular neuron and then mark all inputs whose activation values do not follow such distributions as potentially biased or poisoning samples. The idea of finding a benign distribution is that inputs whose activation values in a layer are non-linear will be considered as benign and the distribution that describes their activation values is used as the reference distribution. In Algorithm 1, line 13 uses a function to test the linearity of activation values and separate all activation values into a benign set $B$ and others $O$. Then, we normalize the distribution of our reference distribution $B$ to a normal distribution and obtain its mean value $\mu$ and standard deviation value $\sigma$ (line 14). For a single input $i$, we perform a statistical test to see the probability of $i$ being a potentially biased or poisoning sample (lines 15 to 16). We exclude it from the training dataset (line 17). The last step of our training algorithm is to remove the compromised neuron effects from the model by resetting them. We reuse the initialization method in our training setting to set a new value for all identified compromised neurons (line 19 and 20). Then, we continue the training

until terminating conditions are met, or the training budget is used up (line 2). This algorithm uses ReLU as an example, and because our theory still holds for other activation functions (see § 8.2), the algorithm can generalize to other models by modifying corresponding parameters (line 9).

To identify compromised neurons and biased/poisoning images, we need to determine the threshold value $\theta$ (line 9) and separate the activation values into non-linear and linear ones. In our implementation, we perform the Fisher's linear discriminant analysis (its binary version, Otsu's method [46]) and leverage the Jenks natural breaks optimization algorithm to find the separations of non-linear and linear activation values. Sets $B$ and $O$ are outputs of such algorithms, and we use the standard value that evaluates the quality of such separation to compute the value of our threshold $\theta$, i.e., 0.95 in our case. Notice such values can affect the accuracy of identifying compromised neurons and inputs. We also choose alternative ways to separate the sets and present the results in the Appendix to evaluate our approach. Similar to model pruning and fine-tuning, when we reset different numbers of neurons (lines 19 and 20), the accuracy on benign and Trojan samples can be different. We present more details of the algorithm and evaluate the sensitivity of NONE to the number of neurons reset during training and identify malicious samples in Appendix (§ 8.8).

## 5    Experiments

NONE is implemented in Python 3.8 with PyTorch 1.7.0 and CUDA 11.0. If not specified, all experiments are done on a Ubuntu 18.04 machine equipped with six GeForce RTX 6000 GPUs, 64 2.30GHz CPUs, and 376 GB memory. We first introduce the experiment setup (§ 5.1). Then, we evaluate the overall effectiveness of NONE (§ 5.2) and investigate its robustness against different attack settings (§ 5.3). We also evaluate the generalization of NONE on real-world applications (§ 5.4). Finally, we measure the precision and the recall in the poisoned samples identification stage (§ 5.5). Other results and discussions are in Appendix.

### 5.1    Experiment Setup.

**Datasets and Models.** We evaluate NONE on five publicly available datasets: MNIST [47], GT-SRB [48], CIFAR-10 [49], ImageNet-10 [50] and TrojAI [51]. The overview of our datasets and more details can be found in § 8.5. Fig. 3 illustrates different trigger patterns used in the experiments (i.e., a single pixel located in the right bottom corner of Fig. 3(a), a fixed red patch in Fig. 3(b), a black-and-white pattern whose location is random in Fig. 3(c) and a colorful watermark in Fig. 3(d)). The default trigger pattern for MNIST, CIFAR-10, GTSRB, and ImageNet-10 are Single Pixel (Fig. 3(a)), Static Patch (Fig. 3(c)), Dynamic Patch (Fig. 3(b)) and Watermark (Fig. 3(d)), respectively. Besides the default triggers for each dataset, we also measure the impacts of using other triggers. The results are included in § 5.3. We evaluate NONE and other defense methods on AlexNet [50], NiN (Network in Network) [19], VGG11, VGG16 [52] and ResNet18 [53]. These models are representative and are commonly used in existing Trojan related studies [8, 26].

**Evaluation Metrics.** We use benign accuracy (BA) and attack success rate (ASR) [54] as evaluation metrics, which is a common practice [55, 41, 26]. BA is defined as the number of correctly classified benign samples over the number of all benign samples. It implies model's capability on its original task. ASR evaluates the success rate of backdoor attacks. It is calculated as the number of backdoor samples that can successfully attack the model over the number of all generated backdoor samples.

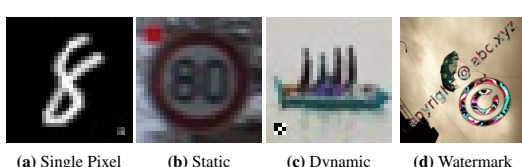

**(a)** Single Pixel    **(b)** Static    **(c)** Dynamic    **(d)** Watermark

**Fig. 3:** Examples of Using Different Trigger Patterns.

**Attack Settings.** As we introduced in § 2, Trojans are classified into two categories: *Injected Trojans* and *Natural Trojans*. We implement both of them to evaluate NONE and other defense methods. For Injected Trojans, we implement both single target and label specific BadNets [1], label-consistent Trojan attack [11] and hidden trigger Trojan attack [56]. Following the original paper, we use images from a pair of classes (class theater curtain and class plunger) in ImageNet for hidden trigger Trojan attack. For natural Trojans, we follow the previous work [8] to reproduce the attack. Due to the fact that we do not know if the data is poisoned or not in advance, NONE keeps detecting Trojan attacks and training the model even if there is no attack activity. We evaluate the

**Table 1:** Comparisons on Injected Trojans.

| Attack Type | Dataset | Network | Undefended | | DP-SGD | | NAD | | AC | | ABL | | NONE | |
|---|---|---|---|---|---|---|---|---|---|---|---|---|---|---|
| | | | BA | ASR | BA | ASR | BA | ASR | BA | ASR | BA | ASR | BA | ASR |
| BadNets (single target) | MNIST | NiN | 99.65% | 99.96% | 92.94% | 0.48% | 99.26% | 0.06% | 98.47% | 0.51% | 98.25% | 99.70% | **99.58%** | **0.06%** |
| | | VGG11 | 99.35% | 100.00% | 97.20% | 1.61% | 98.85% | 2.12% | 97.44% | 10.93% | 98.48% | 0.64% | **99.11%** | **0.14%** |
| | | ResNet18 | 99.61% | 99.98% | 97.31% | 0.13% | 98.83% | 0.12% | 98.04% | 0.25% | 99.34% | **0.04%** | 99.57% | 0.37% |
| | CIFAR-10 | NiN | 90.52% | 100.00% | 36.41% | 93.01% | 80.99% | 52.97% | 83.77% | 100.00% | **90.13%** | 100.00% | 90.11% | **2.32%** |
| | | VGG16 | 90.46% | 100.00% | 55.61% | 99.32% | 88.70% | 98.71% | 88.14% | 99.84% | **89.96%** | 100.00% | 89.70% | **4.91%** |
| | | ResNet18 | 94.10% | 100.00% | 52.29% | 99.99% | 88.74% | 1.28% | 89.57% | 57.43% | 92.40% | 1.66% | **93.62%** | **1.07%** |
| | GTSRB | NiN | 95.95% | 99.72% | 31.54% | 74.78% | **96.53%** | 26.15% | 95.36% | 99.52% | 96.14% | 99.54% | 95.51% | **0.87%** |
| | | VGG16 | 95.43% | 99.93% | 54.60% | 86.83% | **95.94%** | 86.10% | 91.28% | 5.28% | 94.06% | 95.79% | 94.66% | **0.96%** |
| | | ResNet18 | 96.67% | 99.84% | 55.73% | 90.88% | 95.95% | 13.18% | 94.64% | 98.57% | 96.24% | 0.93% | **96.39%** | **0.76%** |
| | ImageNet-10 | NiN | 82.17% | 99.64% | 21.22% | 65.21% | 76.99% | 92.61% | 74.32% | 92.41% | **81.35%** | 99.92% | 80.31% | **0.14%** |
| | | VGG16 | 88.97% | 100.00% | 18.06% | 44.62% | 84.03% | 10.93% | 81.58% | 100.00% | 79.16% | 100.00% | **87.11%** | **0.14%** |
| | | ResNet18 | 89.83% | 99.07% | 40.31% | 29.58% | 86.29% | 25.89% | 82.39% | 98.44% | 83.72% | 6.83% | **86.34%** | **0.08%** |
| BadNets (label specific) | MNIST | NiN | 99.64% | 98.86% | 93.02% | 1.10% | 99.24% | 0.09% | 99.60% | 0.03% | 98.40% | 89.32% | **99.42%** | **0.03%** |
| | | VGG11 | 99.05% | 98.99% | 97.17% | 0.41% | 99.06% | 32.06% | 97.59% | 0.10% | **99.21%** | 98.51% | 98.63% | **0.11%** |
| | | ResNet18 | 99.57% | 99.49% | 96.81% | 0.38% | 99.19% | 0.19% | **99.35%** | **0.03%** | 99.01% | 98.81% | 99.09% | 0.20% |
| | CIFAR-10 | NiN | 90.50% | 78.56% | 38.42% | 6.43% | 82.70% | 13.65% | 84.74% | 56.57% | **89.66%** | 77.69% | 89.51% | **1.27%** |
| | | VGG16 | 90.73% | 96.86% | 55.12% | 5.23% | 88.65% | 39.99% | 88.18% | 82.87% | **90.13%** | 86.91% | 89.64% | **1.22%** |
| | | ResNet18 | 94.37% | 92.00% | 52.19% | 12.17% | 88.21% | 1.50% | 92.68% | 5.29% | 83.83% | 83.48% | **93.05%** | **1.04%** |
| | GTSRB | NiN | 96.06% | 93.74% | 24.70% | 7.39% | **96.46%** | 9.90% | 94.02% | 6.17% | 96.14% | 99.54% | 95.99% | **0.96%** |
| | | VGG16 | 95.71% | 94.57% | 53.90% | 6.71% | **96.33%** | 81.43% | 94.96% | 69.22% | 94.32% | 80.21% | 95.49% | **1.65%** |
| | | ResNet18 | 96.93% | 97.40% | 60.59% | 6.61% | 95.49% | 11.13% | 95.74% | 1.16% | 90.24% | 93.93% | **96.63%** | **0.91%** |
| | ImageNet-10 | NiN | 82.65% | 66.37% | 23.44% | 10.06% | 76.31% | 23.31% | 78.85% | 14.39% | **80.20%** | 56.82% | 79.39% | **6.98%** |
| | | VGG16 | 89.04% | 76.20% | 17.94% | 10.80% | 80.21% | 22.17% | 81.22% | 59.62% | 78.34% | 53.68% | **84.51%** | **8.31%** |
| | | ResNet18 | 88.87% | 59.75% | 44.59% | 7.95% | 85.76% | 15.82% | 79.44% | 23.41% | 81.30% | 50.52% | **84.74%** | **6.55%** |
| Label consistent | CIFAR-10 | NiN | 91.32% | 98.98% | 38.83% | 5.21% | 82.35% | 65.63% | 83.92% | 95.96% | 89.95% | 95.28% | **90.11%** | **2.19%** |
| | | VGG16 | 90.97% | 98.41% | 53.76% | 9.24% | 88.88% | 92.71% | 88.46% | 7.57% | 90.38% | 94.94% | **90.07%** | **4.26%** |
| | | ResNet18 | 94.73% | 83.42% | 54.00% | 11.00% | 90.74% | 60.74% | 88.00% | 65.86% | 87.34% | 2.17% | **94.01%** | **2.14%** |
| Hidden Trigger | ImageNet-pair | AlexNet | 93.00% | 82.00% | 80.00% | 62.00% | 91.00% | 74.00% | 90.00% | 22.00% | 90.00% | 54.00% | **91.00%** | **4.00%** |

**Table 2:** Comparisons on Natural Trojan.

| Dataset | Network | Undefended | | DP-SGD-1 | | DP-SGD-2 | | NONE | |
|---|---|---|---|---|---|---|---|---|---|
| | | BA | ASR | BA | ASR | BA | ASR | BA | ASR |
| CIFAR-10 | NiN | 91.02% | 87.62% | 60.22% | 98.85% | 39.19% | 87.22% | **86.94%** | **34.21%** |
| | VGG16 | 90.78% | 71.88% | 78.25% | 61.11% | 53.40% | 63.58% | **81.83%** | **37.49%** |
| TrojAI | VGG11 | 99.88% | 72.09% | 84.51% | 88.75% | 6.06% | 88.55% | **99.04%** | **56.68%** |
| | Resnet18 | 99.91% | 54.33% | 78.88% | 61.77% | 52.81% | 58.94% | **98.13%** | **34.98%** |

additional costs of NONE on benign models in such non-attack settings. More details for our attack settings are included in § 8.6. Besides above attacks, We also evaluate the generalization of NONE on more attacks in § 8.7.

**Comparison.** We compare NONE with 4 state-of-the-art defense methods: DP-SGD [24], Neural Attention Distillation (NAD) [38], Activation Clustering (AC) [22] and Anti-backdoor Learning (ABL) [9]. We use their official code and default hyperparameters specified in the original papers.

## 5.2 Effectiveness of NONE

**Experiments.** We measure the effectiveness of NONE by comparing the BA and ASR of models protected by NONE with those of undefended models and models protected by existing defense methods. The comparison results on injected Trojans, natural Trojans and non-attack settings are shown in Table 1, Table 2 and Table 3, respectively. In each table, we show the detailed settings including attack settings, dataset names and network architectures., etc. For the evaluation results on natural Trojans ( Table 2), the ASR and BA are the average results under different trigger size settings: 2%, 4%, 6%, 8%, 10% and 12% of the whole image. Notice that, to the best of our knowledge, there is no defense method designed for natural Trojans. We observe that DP-SGD can potentially mitigate natural Trojans because it reduces the high gradients brought from natural Trojans. Therefore, we adapt DP-SGD as the baseline method for natural Trojans. We configure DP-SGD with two settings of parameters following the prior work [24]. For DP-SGD-1, we set the clip as 4.0 and the noise as 0.1. The clip and noise of DP-SGD-2 are 1.0 and 0.5. For the non-attack settings, we deploy NONE on several benign models and show the decrease of BA in Table 3.

**Table 3:** Benign Accuracy in Non-attack Settings.

| Dataset | Network | Without NONE | With NONE |
|---------|---------|--------------|-----------|
| CIFAR-10 | NiN | 91.02% | 89.40% |
| | VGG16 | 90.78% | 89.62% |
| | ResNet18 | 94.83% | 93.92% |
| GTSRB | NiN | 95.68% | 95.36% |
| | VGG16 | 94.67% | 94.08% |
| | ResNet18 | 96.89% | 96.87% |
| ImageNet-10 | NiN | 83.34% | 79.18% |
| | VGG16 | 88.84% | 83.41% |
| | ResNet18 | 89.81% | 85.25% |

**Table 4:** Results on Different Trigger Patterns.

| Trigger Pattern | Network | Undefended | | NONE | |
|-----------------|---------|------|------|------|------|
| | | BA | ASR | BA | ASR |
| Dynamic Patch | NiN | 90.52% | 100.00% | 90.11% | 2.32% |
| | VGG16 | 90.46% | 100.00% | 89.70% | 4.91% |
| | ResNet18 | 94.10% | 100.00% | 93.62% | 1.07% |
| Static Patch | NiN | 90.92% | 100.00% | 89.93% | 2.61% |
| | VGG16 | 90.12% | 100.00% | 89.48% | 4.03% |
| | ResNet18 | 94.24% | 99.99% | 93.93% | 1.37% |
| Watermark | NiN | 90.88% | 99.99% | 87.74% | 3.27% |
| | VGG16 | 90.64% | 99.99% | 89.14% | 5.36% |
| | ResNet18 | 94.28% | 100.00% | 92.27% | 5.99% |

**Table 5:** Results on Different Trigger Sizes.

| Trigger Size | Undefended | | NONE | |
|--------------|------|------|------|------|
| | BA | ASR | BA | ASR |
| 3*3 | 94.10% | 100.00% | 93.62% | 1.07% |
| 5*5 | 94.27% | 100.00% | 93.51% | 1.34% |
| 7*7 | 94.10% | 99.98% | 93.70% | 1.53% |
| 9*9 | 94.34% | 100.00% | 93.19% | 5.18% |
| 15*15 | 94.44% | 100.00% | 92.18% | 32.27% |
| 20*20 | 94.58% | 100.00% | 92.55% | 99.82% |

**Table 6:** Results on Different Poisoning Rates.

| Poisoning rate | Undefended | | NONE | |
|----------------|------|------|------|------|
| | BA | ASR | BA | ASR |
| 0.50% | 94.46% | 100.00% | 93.50% | 2.52% |
| 5.00% | 94.10% | 100.00% | 93.62% | 1.07% |
| 10.00% | 93.82% | 100.00% | 93.13% | 1.04% |
| 20.00% | 92.70% | 100.00% | 92.14% | 1.39% |

**Results on Injected Trojans.** From the results on Injected Trojan attacks (Table 1), we observe that applying NONE can better protect models from being attacked by injected Trojans than other defense methods. With NONE, the average ASR of models decreases from 93.34% to 1.91%, which is much better than other defense methods (DP-SGD, NAD, AC and ABL can only reduce the average ASR to 30.32%, 34.08%, 45.45% and 68.60% respectively). The reason is that, unlike existing methods based only on specific empirical observations, NONE targets the root cause of the Trojans (i.e., the linearity) and reveals the attacks more accurately. Therefore, NONE can better defend against attacks than other methods.

We also find that NONE almost does not have negative impacts on the original task of models. From Table 1, the BA of NONE is the highest among all methods and is similar to that of undefended models, meaning that NONE has a low defense cost. The reason is that NONE only modifies the compromised neurons that are highly relevant to Trojan but less related to the original task of models. Therefore, most of the benign knowledge is preserved when applying NONE and the model can still perform well on its original tasks. Meanwhile, NONE finetunes the model on the purified data after the reset process, further strengthening the capabilities of models and reducing defense costs.

It is worth clarifying that ABL has a poor performance on label-specific attacks and other attacks that use NiN and VGG models. The possible reason is that the design of ABL requires the model to learn quicker and better on Trojan samples than benign samples [9] (i.e., the learning of Trojan samples should have a lower training loss value in the early training stage). When the attacker uses more complex attacks (e.g., label specific BadNets) or models with limited learning capabilities (e.g., NiN and VGG), the model cannot learn Trojan samples quickly, leading to poor performance.

**Results on Natural Trojans.** From the results in Table 2, we find that NONE protects the model most effectively and has the lowest defense costs among all defense methods. Overall, applying NONE achieves 1.75 times lower ASR than undefended models. DP-SGD methods can only slightly decreases ASR or even increase ASR. The results show that using NONE is the most effective way for natural Trojan defense. The loss of BA using NONE is also smaller than the loss caused by applying the DP-SGD methods (3.90% with NONE and 38.76% with DP-SGD methods on average), further showing the efficiency of NONE. The results confirm our analysis: reducing the linearity of models reduces the ASR of natural Trojans without posing much additional cost.

**Results on Non-attack Setting.** We also explore whether NONE affects the performance of benign models on their original tasks. From Table 3, we find that NONE has a low effect on benign models. Applying NONE only decreases 2.33% BA on average. Because only a few neurons are detected as compromised neurons and are reset by NONE when there is no Trojan activity, NONE does not affect the learned benign knowledge. Moreover, the subsequent training process further reduces the costs. Therefore, we conclude that NONE does not impose high additional costs on benign models.

**Table 7:** Results on Federated Learning.

| Dataset | Undefended | | NONE | |
|---|---|---|---|---|
| | BA | ASR | BA | ASR |
| MNIST | 99.22% | 99.15% | 98.60% | 0.13% |
| CIFAR-10 | 80.31% | 43.23% | 78.67% | 4.44% |

## 5.3 Robustness of NONE

We evaluate the robustness of NONE against various attack settings (e.g., different trigger sizes, trigger patterns and poisoning rates). If not specified, the model used in the evaluation is ResNet18. The dataset is CIFAR-10 and the evaluated attack is the single target BadNets attack.

**Trigger Sizes.** To study the effects of trigger size, we use triggers of different sizes (from 3*3 to 20*20) to attack models and collect the ASR and BA of applying NONE on these compromised models. The results are shown in Table 5. Overall, the BA of undefended models and models protected by NONE is insensitive to the change of trigger size. The difference between the highest and lowest BA on the unprotected and protected models is 0.48% and 1.52%, respectively, which is very small. We believe that the BA does not change significantly because triggers usually do not affect the learning of benign features used for original tasks, as discussed in previous work [1].

On the other hand, the trigger size affects the ASR of protected models. When the trigger size becomes larger, the ASR of the protected model increases dramatically from 1.07% to 99.82% and NONE fails. The results are understandable because a large trigger size modifies more pixels in the original image, making the triggers obvious and easy to learn. When the trigger is large, it almost covers the whole image and becomes the majority of the image. In such a scenario, models easily capture trigger features and are compromised. A detailed example is shown in Fig. 10. Currently, the sensitivity to trigger sizes is a common limitation for Trojan defense methods [26, 8]. Considering that a large trigger size is almost impractical because it makes the trigger too obvious to be detected directly by administrators, we consider NONE robust to most trigger sizes.

**Trigger Patterns.** To measure the robustness of NONE against different attack trigger patterns, we use NONE to protect models from being attacked by Dynamic Patch trigger, Static Patch trigger and Watermark trigger. The results are shown in Table 4. We observe that NONE always achieves low ASR and high BA under different trigger settings. The results demonstrate that NONE is effective against different trigger pattern settings. Moreover, we notice that the ASR of using the watermark trigger is particularly larger compared with using other triggers. The reason is that the watermark triggers are large and more complex, as shown in Fig. 3.

**Poisoning Rates.** To measure the impacts of different poisoning rates, we collect the ASR and BA of models being compromised at different poisoning rates from 0.50% to 20.00%. The results are summarized in Table 6. Based on the results, we find that increasing the poisoning rate slightly decreases both the BA and ASR. Specifically, the BA of the model decreases by 1.36% and the ASR of the model decreases by 1.13%. This is because increasing the poisoning rate reduces the number of benign samples used for training, and the BA of the model naturally decreases. Meanwhile, a large poisoning rate makes Trojans easy to be detected, which leads to a lower ASR. Since the changes in ASR and BA are quite small, NONE is considered robust to most poisoning rates.

## 5.4 Generalization on Complex Applications

Attackers may conduct attacks in a more complex scenario. To measure the generalization of NONE on complex applications, we evaluate NONE on two federated learning applications trained on different datasets (i.e., MNIST and CIFAR-10) and a transfer learning application. Each federated learning application has 10 participants, of which 4 of them are malicious participants who conduct the distributed Trojan attacks [7] jointly to inject Trojan triggers into the global model. We assume that attackers train their local models on the poisoned training data and contribute to the global model without scaling the original weight of the poisoned local models. We then apply NONE on the global model to defend against the Trojan attack from malicious local models. Specifically, NONE requires the participants to use their data to test the global model and upload the activation values of the global model to identify compromised neurons. To measure the defense performance,

**Table 8:** Precision and Recall of Poisoned Samples Identification.

| Attack | NiN | | VGG16 | | ResNet18 | |
|---|---|---|---|---|---|---|
| | Precision | Recall | Precision | Recall | Precision | Recall |
| BadNets | 99.60% | 99.64% | 99.96% | 100.00% | 99.84% | 99.92% |
| Label-consistent | 98.80% | 99.20% | 100.00% | 100.00% | 100.00% | 100.00% |

we measure the BA and ASR of the global models (i.e., the original model and model deployed with NONE). The results are shown in Table 7. As shown in the table, on average, NONE achieve 31.16 times lower ASR than undefended models with a slight decrease (i.e., 1.13%) in the BA. The results show that NONE can defend against the Trojan attacks effectively in real-world federated learning applications at a low cost. Besides federated learning, we have also discussed the performance of NONE in transfer learning settings. Hidden Trigger Trojan Attack [56] in § 5.2 of the main paper is conducted in transfer learning scenarios and the results are shown in Table 1 of the main paper. As the results show, NONE achieves low ASR (i.e., 4.00%) and high BA (i.e., only 2.00% lower than undefended models), proving the generalization of NONE on transfer learning settings.

### 5.5 Precision and Recall of the Poisoned Sample Identification

In line 12-17 of Algorithm 1, we identify the poisoned samples in the training data. To evaluate the effectiveness of the identification process, we measure the precision and the recall of detecting poisoned samples on the CIFAR-10 dataset and three different models (i.e., NiN, VGG16, and ResNet18). Results in Table 8 demonstrate the precision and the recall of NONE are always above 98% on different settings. For example, on ResNet18 and Label-consistent attack, both the precision and the recall of NONE are 100.00%. Thus, NONE can detect poisoned samples accurately.

## 6   Discussion

In this paper, we focus the discussion on image classification tasks, which is the focus of many existing works [1, 56, 11, 8, 26, 57]. Expanding our work, including the theory and system to other problem domains, such as natural language processing and reinforcement learning, other computer vision tasks, e.g., object detection, will be our future work.

Research on adversarial machine learning potentially has ethical concerns. In this research, we propose a theory to explain existing phenomena and attacks, and propose a new training method that removes Trojans in a DNN model. We believe this is beneficial to society.

In our current threat model, the adversary can only inject poisoning data into the training dataset, and there is no existing adaptive attacks. However, adaptive adversaries can still conduct attacks under other threat models, and we discuss such case in § 8.10.

## 7   Conclusion

In this paper, we present an analysis on DNN Trojans and find relationships between decision regions and Trojans with a formal proof. Moreover, we provide empirical evidence to support our theory. Furthermore, we analyzed the reason why models will have such phenomena is because of linearity of trained layers. Based on this, we propose a novel training method to remove Trojans during training, NONE, which can effectively and efficiently prevent intended and unintended Trojans.

### Acknowledgement

We thank the anonymous reviewers for their valuable comments. This research is supported by IARPA TrojAI W911NF-19-S-0012. Any opinions, findings, and conclusions expressed in this paper are those of the authors only and do not necessarily reflect the views of any funding agencies.

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
