# 8 Appendix

**Roadmap:** In this appendix, we first list the symbols we used in this paper. We show the proof and the empirical results for Theorem 3.3 (§ 8.1), more empirical evidence for model's linearity (§ 8.2), explanation for DP-SGD (§ 8.3), and more details about the sample separation (§ 8.4). Then, we provide implementation details including details of datasets (§ 8.5) and used attacks (§ 8.6). We then discuss the the resistance of NONE against more attacks (§ 8.7), and the efficiency of NONE (§ 8.9). We measure the sensitivity of NONE against different configurable parameters (§ 8.8). We then evaluate NONE on an adaptive attack (§ 8.10). In § 8.11, we compare NONE to another defense DBD [58]. We also compare NONE to more defenses on natural Trojan in § 8.12. Finally, we evaluate the generalization to larger models (§ 8.13) and larger datasets (§ 8.14).

**Symbol Table.**

**Table 9:** Summary of Symbols

| Scope | Symbol | Meaning |
|---|---|---|
| Theory | $\boldsymbol{x}$ | Benign Sample |
| | $\tilde{\boldsymbol{x}}$ | Trojan Sample |
| | $T$ | Trojan Sample Generation Function |
| | $\boldsymbol{m}$ | Mask of Trojan Trigger |
| | $\boldsymbol{t}$ | Pattern of Trojan Trigger |
| | $\odot$ | Hadamard product |
| | $\mathcal{M}$ | Model |
| | $\mathcal{X}$ | Input Domain |
| | $\mathcal{Y}$ | Set of Labels |
| | $\mathcal{R}^l$ | Decision Region of Label $l$ |
| | $\mathcal{T}$ | Trojan Region |
| | $m$ | The Number of Elements in $\boldsymbol{m}$ |
| | $\{\boldsymbol{Ax} - \boldsymbol{b} = 0\}$ | Trojan Hyperplane |
| | $x_j$ | Inputs of Layer $j$ |
| | $y_j$ | Outputs of Layer $j$ |
| | $\mathbf{W}_j$ | Trained Weights of Layer $j$ |
| | $\mathbf{b}_j$ | Trained Bias of Layer $j$ |
| Algorithm | $D$ | Training Data |
| | $E$ | Maximal Epoch |
| | $e$ | Current Epoch |
| | $M$ | Model |
| | $n$ | Neuron |
| | $A$ | Activation Values |
| | $A_n$ | Activation values on Neuron $n$ |
| | $C$ | Compromised Neurons |
| | $B_n$ | The Cluster of Smaller Values in $A_n$ |
| | $O_n$ | The Cluster of Larger Values in $A_n$ |
| | $\mu$ | Mean Value of $B_n$ |
| | $\sigma$ | Standard Deviation Value of $B_n$ |
| | $i$ | Input Sample |
| | $i_n$ | The Activation Value of Input Sample $i$ on Neuron $n$ |

## 8.1 Proof and Empirical Evidence for Theorem 3.3

We start our analysis from ideal Trojan attacks, which we define as complete and precise Trojans:

**Definition 8.1. Complete Trojan**: For a Trojaned model $\mathcal{M} : \mathcal{X} \mapsto \mathcal{Y}$ with trigger $(\boldsymbol{m}, \boldsymbol{t})$ and target label $l$, we say a Trojan is complete if $\forall \boldsymbol{x} \in T(\mathcal{X}, \boldsymbol{m}, \boldsymbol{t}), \mathcal{M}(\boldsymbol{x}) = l$.

**Definition 8.2. Precise Trojan**: For a Trojaned model $\mathcal{M} : \mathcal{X} \mapsto \mathcal{Y}$ with trigger $(\boldsymbol{m}, \boldsymbol{t})$ and target label $l$, we say a Trojan is precise if the follow condition is met: $\forall (\boldsymbol{m}', \boldsymbol{t}') \neq (\boldsymbol{m}, \boldsymbol{t}), \boldsymbol{x}' = T(\boldsymbol{x}, \boldsymbol{m}', \boldsymbol{t}'), \mathcal{M}(\boldsymbol{x}) \neq l \Rightarrow \mathcal{M}(\boldsymbol{x}') \neq l$.

Intuitively, a complete Trojan means the attack success rate of this attack is 100%, and a precise Trojan means that the trigger is unique: if we change the trigger ($\boldsymbol{t}$ or $\boldsymbol{m}$), it will not trigger the predefined misclassification.

*Proof.* In Theorem 3.3, we have $\mathbf{S}_0 \iff \mathbf{S}_1$ where:

- $\mathbf{S}_0$: Trojan in $\mathcal{M}$ with trigger being $(\boldsymbol{m}, \boldsymbol{t})$ and target label being $l$ is a complete and precise Trojan.

- **S₁**: The hyperplane $\{Ax - b = 0\}$ is the Trojan region of $\mathcal{M}$ and the only one, where $i \in \{1 \dots m\}$, diagonal matrix $A_{i,i} = m_i, b = At$.

In this proof, we first prove $x \in T(\mathcal{X}, m, t) \iff x \in \{Ax - b = 0\}$, and then prove $\mathbf{S}_0 \Rightarrow \mathbf{S}_1$ and $\mathbf{S}_1 \Rightarrow \mathbf{S}_0$.

**Step 1:** $x \in T(\mathcal{X}, m, t) \Rightarrow x \in \{Ax - b = 0\}$. Let $\tilde{x}$ be a Trojan input generated from $x$ by applying Eq. 1 (in § 3 of the main paper), $\tilde{x} = T(x, m, t) = (1 - m) \odot x + m \odot t$, we get:

$$A\tilde{x} - b = A((1 - m) \odot x + m \odot t) - b \tag{2}$$

Then, based on the definition of matrix $A$, $b$, the Hadamard product, and the distributive property of matrix multiplication, we can get the following equation, where $E$ is the identity matrix:

$$\begin{aligned} A((1 - m) \odot x + m \odot t) - b &= A((E - A)x + At) - At \\ &= A(E - A)x + AAt - At \end{aligned} \tag{3}$$

Since $A$ is a diagonal matrix, and all elements of $A$ is 0 or 1 based on the definition of $A$ and trigger mask $m$, we can get $AA = A$ and $A(E - A) = 0$. Then, according to Eq. 2 and Eq. 3, we get: $\forall x \in T(\mathcal{X}, m, t), \ Ax - b = 0$.

**Step 2:** $x \in \{Ax - b = 0\} \Rightarrow x \in T(\mathcal{X}, m, t)$. This step is to prove that any sample in the hyperplane $\{Ax - b = 0\}$ can be obtained from pasting Trojan trigger on other samples. Let $\tilde{x}$ denote any sample in the hyperplane, and $x$ is the sample that is not in the hyperplane, i.e., an external sample. Any external sample $x$ that satisfies $(E - A)x = (E - A)\tilde{x}$ can be transformed to $\tilde{x}$ via the projection specified by $m$ and $t$. Therefore, we conclude that any sample in the hyperplane can be obtained by pasting the Trojan trigger on other samples.

Steps 1 and 2 prove that $x \in T(\mathcal{X}, m, t)$ is equivalent to $x$ in the hyperplane $Ax - b = 0$, namely:

$$x \in T(\mathcal{X}, m, t) \iff x \in \{Ax - b = 0\} \tag{4}$$

**Step 3:** $\mathbf{S}_0 \Rightarrow \mathbf{S}_1$. Based on $\mathbf{S}_0$, Trojan in $\mathcal{M}$ is a complete Trojan. Based on Eq. 4 and the definition of complete Trojan (i.e., Theorem 8.2), we get: $\forall x \in \{Ax - b = 0\}, \mathcal{M}(x) = l$, which means $\{Ax - b = 0\}$ is a Trojan decision region. We then prove $\{Ax - b = 0\}$ is the only Trojan region using proof by contradiction. For any other hyperplane $\{A'x - b' = 0\}$ where $(A', b') \neq (A, b)$, based on Eq. 4, we can get: $(m', t') \neq (m, t), x' = T(x, m', t') \iff A'x' - b' = 0$. According to $\mathbf{S}_0$, the Trojan is a precise Trojan: $\forall (m', t') \neq (m, t), x' = T(x, m', t'), \mathcal{M}(x) \neq l \Rightarrow \mathcal{M}(x') \neq l$. Thus, we get that $A'x - b' = 0$ is not a Trojan region. That is, the Trojan region has only one hyperplane, $\{Ax - b = 0\}$.

**Step 4:** $\mathbf{S}_1 \Rightarrow \mathbf{S}_0$. According to $\mathbf{S}_1$, we have:

$$\forall x \in \{Ax - b = 0\}, \mathcal{M}(x) = l \tag{5}$$

$$\forall (A', b') \neq (A, b), A'x' - b' = 0, \mathcal{M}(x) \neq l \Rightarrow \mathcal{M}(x') \neq l \tag{6}$$

From Eq. 4 and Eq. 5, we get $\forall x \in T(\mathcal{X}, m, t), \mathcal{M}(x) = l$, which means the Trojan is complete. Based on Eq. 4 and Eq. 6, we can get $\forall (m', t') \neq (m, t), x' = T(x, m', t'), \mathcal{M}(x) \neq l \Rightarrow \mathcal{M}(x') \neq l$, where $m' = A'_{i,i}, b' = A't'$, indicating that the Trojan is precise.

From Step 3 and 4, we can conclude that $\mathbf{S}_0 \iff \mathbf{S}_1$, and complete the proof of Theorem 3.3. ☐

Intuitively, the Trojan is precise means the attack success rate is 100% which guarantees that all samples with the trigger will be classified as the target label. The Trojan is complete means that no other input patterns can trigger this trigger, and thus all inputs that activate this Trojan have this trigger. In the real world, these are hard to achieve. In practice, a Trojan of model $\mathcal{M}$ whose trigger is $(m, t)$ and target label is $l$ has

$$\exists (m', t') \approx (m, t), \mathbb{P}(\mathcal{M}(T(x, m', t')) = l) > \lambda \tag{7}$$

$$\mathbb{P}(\mathcal{M}(T(x, m, t)) = l) < 1, x \in \mathcal{D} \tag{8}$$

where $\mathcal{D}$ is the dataset, and $\lambda$ is a threshold value for the attack success rate (e.g., 90%). Namely, in the real world, a Trojan trigger cannot guarantee a 100% attack success rate and the model can

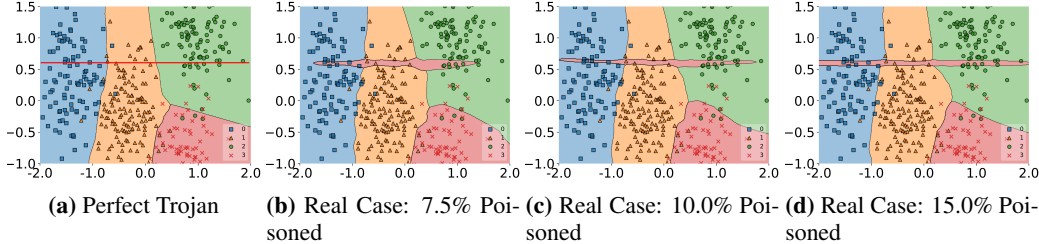

**(a)** Perfect Trojan  **(b)** Real Case: 7.5% Poisoned  **(c)** Real Case: 10.0% Poisoned  **(d)** Real Case: 15.0% Poisoned

**Fig. 4:** Perfect Trojans and Relaxations on 2D Data. Each sub-figure contains test samples (dots) and the learned decision regions for different labels (in different colors) under a specific setting. The Trojan trigger is $t = (-, 0.6)$, and the target label $y_t = 3$. The red region near (-, 0.6) is the learned Trojan decision region.

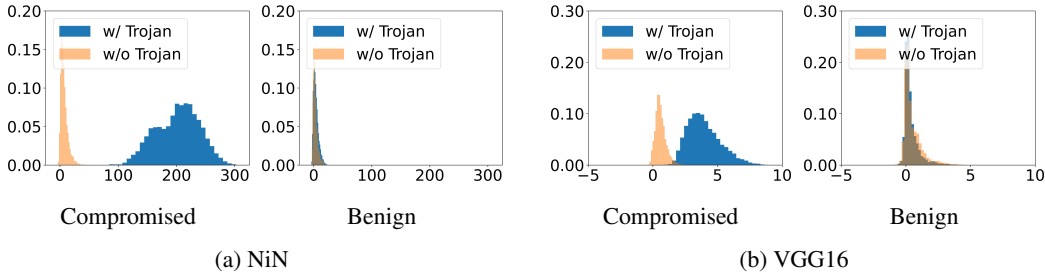

(a) NiN  (b) VGG16

**Fig. 5:** Comparison of Activation Values on Different Network Architectures.

learn a trigger that is different from the intended one. Consequently, the real Trojan region $\mathcal{T}'$ and the theoretical one $\mathcal{T}$ satisfy $\frac{|\mathcal{T}' \cap \mathcal{T}|}{|\mathcal{T}|} = \alpha$ where $\alpha$ is the real attack success rate.

To evaluate if the Trojan decision region in real-world data is the relaxation of the Trojan linear hyperplane, we visualize the decision regions of Trojaned neural networks.

Following Bai et al. [59], we visualize the decision region of neural networks on 2d data. Specifically, We visualize decision regions of compromised Multilayer Perceptrons (MLP) trained on different poisoning rates. The MLP model has 5 layers and each layer contains 100 neurons, and we use ReLU as the activation function. Similar to Bai et al. [59], the used dataset contains five isotropic Gaussian 2d blobs, in which each blob represents a class. In Fig. 4, we show the complete and precise Trojan decision region (Fig. 4(a)) for this model and real-world relaxations with different poisoning rates of BadNets attack (Fig. 4(b), Fig. 4(c), Fig. 4(d)). Each color in the figure denotes one output label. In our experiments, we set the trigger to $t = (-, 0.6)$, and the target class $y_t = 3$ (red). Thus, the red region close to $t = (-, 0.6)$ denotes the Trojan region. We observe that, with the growth of the poisoning ratio, the attacks get a higher attack success rate and become more precise, and the Trojan region also converts to the ideal one shown in Fig. 4(a). Despite such relaxations, we can also confirm that the Trojan region has a large intersection with the hyperplane and other possible triggers are around the ground truth one.

### 8.2 Empirical Evidence for Theorem 3.3 on Other Models

**Different model architectures.** To evaluate the linearity of different model architectures, we collect the activation outputs of models with different architectures (i.e., NiN and VGG16). Similar to § 3 in the main paper, we use both benign samples and compromised samples as the input of models and collect their activation outputs. The results are shown in Fig. 5. The results show that compromised samples always lead to significantly higher activation values than benign samples in different models. The conclusion is consistent with the linearity theory in § 3 of the main paper and proves that our theory can generalize to different model architectures.

**Different model layers:** Besides the linearity on different model architectures, we also evaluate the linearity on different model layers. Fig. 6 demonstrates the activation outputs of different convolutional layers (i.e., 14[th] to 17[th] layers). Note that we only show the results on 4 layers due to the space limitation.The results on other layers are similar. From the results, we observe that Trojans introduce a large set of high activation values in each layer, leading to the final linearity between

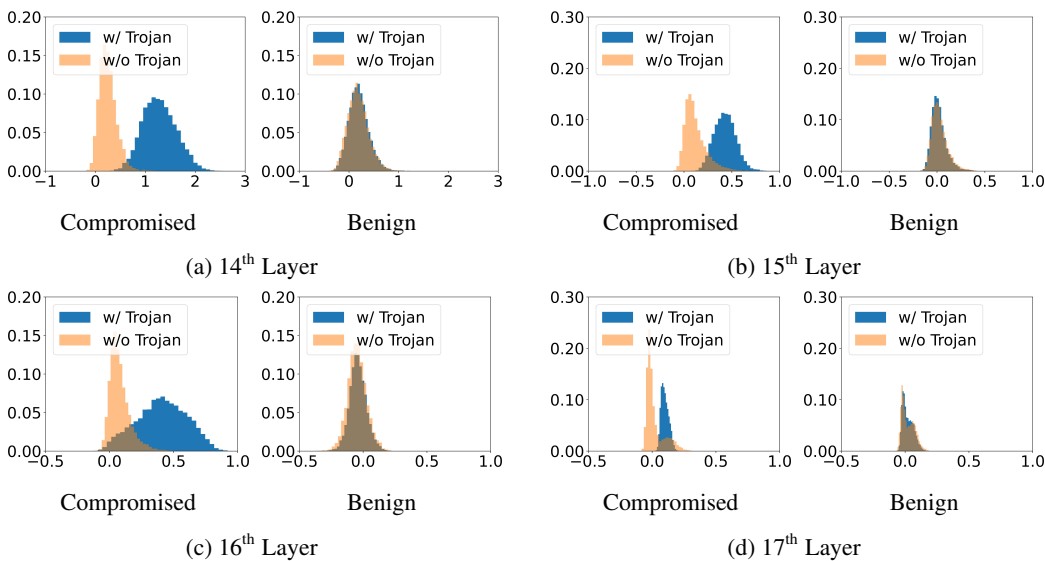

**Fig. 6:** Comparison of Activation Values on Different CNN Layers in ResNet18 Model.

input and activation output. The results are consistent with our previous analysis in § 3 of the main paper and further confirm that Trojans can introduce linearity at each layer of the DNN model.

**Different activation functions.** To investigate if our theory and NONE can generalize on different activation functions, we train 5 ResNet18 models on CIFAR-10 with 2 common used linear activation functions (i.e., ReLU [60], LeakyReLU [61]) and 3 non-linear activation functions (i.e., ELU [62], Tanhshrink [63] and Softplus [64]). Then we apply NONE to protect these models. We report the ASR and BA of both protected models and undefended models. The results are shown in Table 10. Overall, we find that NONE always achieves a low ASR when using different activation functions, showing the generalization of NONE on different activation functions. Even with non-linear activation functions, NONE is still effective and we suspect the reason is that even though some activation functions are non-linear, well-trained deep neural networks do fall into the "highly linear" regions. The results are also consistent with existing papers [44].

**Table 10:** Evaluation Results with Different Activation Functions.

| Activation Function | Undefended | | NONE | |
|---|---|---|---|---|
| | BA | ASR | BA | ASR |
| ReLU | 94.10% | 100.00% | 93.62% | 1.07% |
| LeakyReLU | 94.32% | 100.00% | 93.48% | 1.24% |
| ELU | 92.99% | 99.93% | 91.11% | 1.46% |
| Tanhshrink | 91.68% | 99.76% | 90.18% | 5.11% |
| Softplus | 92.81% | 100.00% | 89.91% | 2.07% |

## 8.3 Explaining DP-SGD Defense

DP-SGD [24] improves existing SGD methods by removing the noises added to poisoning training samples and shadows promising results in defending against Trojans. Here, we explain why it works. Specifically, we use the same settings with Fig. 4 to train 2 compromised models with vanilla SGD and DP-SGD, and show the comparison results in Fig. 7. Results show that data poisoning can successfully attack the vanilla SGD method. As a comparison, DP-SGD makes the decision region (red) much more complex, and removes the malicious "hyperplane" effects to defense against Trojans. Recently, Tursynbek et al. [65] quantitatively measured the curvature of DNN using Curvature Profile [66] and showed that models trained with DP-SGD produce more curved decision boundaries, which is consistent with our results. By doing so, DP-SGD breaks the "hyperplane" Trojans rely on and hence, removes the Trojan effects. However, this unavoidably affects the accuracy of benign samples. As shown in Fig. 7, many benign samples got misclassified.

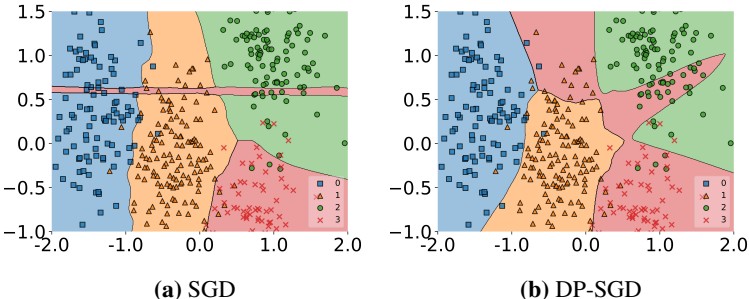

**(a)** SGD            **(b)** DP-SGD

**Fig. 7:** Decision Region Generated by SGD and DP-SGD.

---

**Algorithm 2** Jenks Natural Breaks Optimization

---

**Input:** All Activation Values: $A_n$
**Output:** Cluster of smaller values: $B_n$, Cluster of larger values: $O_n$

  1: **function** SEPARATION($A_n$)
  2:      **for** break $b$ in $Breaks$ **do**
  3:          $B'_n = \{x \in A_n \| x \leq b\}$
  4:          $O'_n = \{x \in A_n \| x \geq b\}$
  5:          $\mu_B, \sigma_B = norm(B'_n)$
  6:          $\mu_O, \sigma_O = norm(O'_n)$
  7:          $\sigma^2_{within} = \sigma^2_B + \sigma^2_O$
  8:          $\sigma^2_{between} = (\mu_B - \mu_O)^2$
  9:          **if** $\sigma_{within}/\sigma_{between} \leq$ lowest **then**
10:             $lowest = \sigma_{within}/\sigma_{between}$
11:             $O_n = O'_n$
12:             $B_n = B'_n$

---

### 8.4 Sample Separation

In line 13 of Algorithm 1, we separate the activation values into two clusters via Fisher's linear discriminant analysis. In detail, we minimize the variance within clusters $\sigma_{within}$ and maximize the variance between clusters $\sigma_{between}$. The process is implemented by Jenks natural breaks optimization, which is an iterative optimization method that finds the minima/maxima of $\sigma_{within}/\sigma_{between}$. The detailed process can be found in Algorithm 2. In line 2 of Algorithm 2, it iterates all possible breaks. In lines 5 to 8, it calculates the value of $\sigma_{within}$ and $\sigma_{between}$. Lines 9 to 12 find the lowest value of $\sigma_{within}/\sigma_{between}$ and the best separation.

### 8.5 Dataset Details

The overview of the dataset is shown in Table 11. Specifically, we order the datasets with their data sizes and show their dataset names, input size of each sample, the total number of samples, the number of classes and the default Trojan triggers used for generating poisoned data in each column. Among these datasets, MNIST [47] is widely used for digit classification tasks. The GTSRB [48] dataset is used for traffic sign recognition tasks in the self-driving scenario. TrojAI [51] contains the images created by compositing a synthetic traffic sign, with a random background image from the KITTI dataset [67]. Other datasets (i.e., CIFAR-10 [49] and ImageNet-10[2]) are built for recognizing general objects (e.g., animals, plants and handicrafts). The default triggers (Fig. 3 in the main paper) used for each dataset are shown in the last column of Table 11. All datasets used in the experiments are with MIT license. They are open-sourced and do not contain any personally identifiable information or offensive content.

---

[2]https://github.com/fastai/imagenette

**Table 11:** Overview of Datasets.

| Name | Input Size | Samples | Classes | Trigger |
|------|-----------|---------|---------|---------|
| MNIST | 28*28*1 | 60000 | 10 | Single Pixel |
| GTSRB | 32*32*3 | 39209 | 43 | Static |
| CIFAR-10 | 32*32*3 | 50000 | 10 | Dynamic |
| ImageNet-10 | 224*224*3 | 9469 | 10 | Watermark |
| TrojAI | 224*224*3 | 125000 | 5-25 | Natural Trojans |

**Table 12:** Results on More Attacks.

| Dataset | Network | Attack | Undefended | | NAD | | ABL | | NONE | |
|---------|---------|--------|------|------|------|------|------|------|------|------|
| | | | BA | ASR | BA | ASR | BA | ASR | BA | ASR |
| CIFAR-10 | ResNet18 | WaNet | 94.39% | 96.71% | 88.81% | 1.17% | 90.79% | 2.68% | **92.24%** | **0.69%** |
| | | SIG | 94.34% | 99.08% | 88.26% | 1.42% | 91.44% | 1.29% | **93.79%** | **1.08%** |
| | | Filter | 91.08% | 99.34% | 87.91% | 4.38% | 88.46% | 2.24% | **89.87%** | **1.20%** |
| | | Blend | 94.62% | 99.86% | 88.24% | 1.58% | 92.72% | 1.70% | **94.21%** | **0.93%** |

## 8.6 Attack Details

We first evaluate the performances of NONE against BadNets [1] on two different settings: single target attack and label specific attack. For the single target attack, we set the label whose index is 0, 0, 1 and 1 as the target label for MNIST, CIFAR-10, GTSRB and ImageNet-10, respectively. For label specific attack, the target label of each sample is the label whose index is (the label index of this sample plus 1)%(the number of classes in the dataset). Then, we evaluate the defense against the label-consistent attack [11] and the natural Trojan attack [8]. We use the same implementation and parameters in original papers to achieve these attacks and compare NONE with other defense methods. Notice that for the label-consistent attack, the official github repository[3] only provides poisoned CIFAR-10 datasets, and the code for training GAN and generating poisoned samples are not released. Therefore, we only evaluate NONE on CIFAR-10. For defending against the hidden trigger Trojan attack [56], we follow the parameter settings in original paper and use a pair of image categories (i.e., randomly selected from ImageNet dataset in the previous work [56]) for testing.

## 8.7 Resistance to More Attacks

In this section, we evaluate the resistance of NONE to more attacks. Four state-of-the-art poisoning based Trojan attacks (WaNet [68], SIG attack [69], Filter attack [8] and Blend attack [10]) are included in the experiments. The dataset and the network used are CIFAR-10 and ResNet18. We report the BA and ASR of undefended model, and the model trained with NAD [38], ABL [9] and NONE. Results in Table 12 demonstrates NONE has better performance than baseline methods (i.e., NAD and ABL). On average, NONE has 0.98% ASR and 92.52% BA. The results indicate that our method is resistance to various Trojan attacks.

## 8.8 Sensitivity to Configurable Parameters

NONE has a few configurable parameters that may affect its performance: learning rate in training, resetting fraction, number of neurons in each layer used to detect malicious samples (selection threshold) and different thresholds used for the identification of compromised neurons. We vary the configurable parameters in NONE independently and evaluate the impact of each. The setting of dataset, models and attack type is the same as evaluation in § 5.3 of the main paper. We use 5% poisoning rate, 3*3 trigger size as the default attack setting.

**Learning rate.** Learning rate usually affects the accuracy and convergence speed of the model during the training process. To understand how the learning rate impacts the model deployed with NONE, we choose learning rates from 0.01 to 0.00001 and then measure the BA and ASR of models using different learning rates in the training process. The results are shown in Fig. 8.

Overall, as shown in Fig. 8(a), using a larger learning rate makes the convergence process faster and the BA lower, except for using the learning rate 0.01. This is because using a larger learning rate can

---

[3]https://github.com/MadryLab/label-consistent-backdoor-code

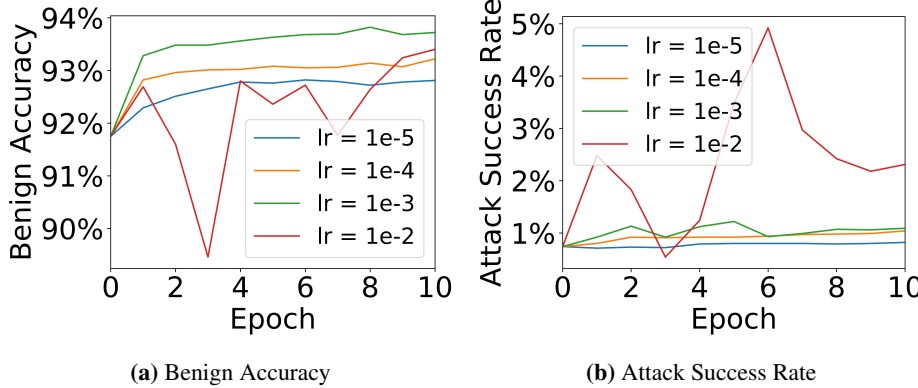

**(a)** Benign Accuracy  **(b)** Attack Success Rate

**Fig. 8:** Evaluation Results with Different Learning Rates.

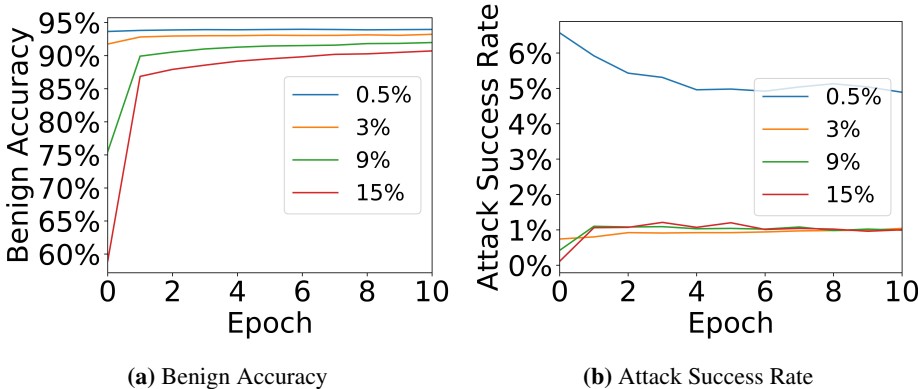

**(a)** Benign Accuracy  **(b)** Attack Success Rate

**Fig. 9:** Evaluation Results with Different Resetting Fractions.

update the weights quickly, but a too large learning rate makes it difficult to find the local optimum and decrease the BA.

In addition, in Fig. 8(b), we find that the final ASR is decreased with the decrease of learning rate after the model is converged. Using learning rate 0.00001 finally achieves the lowest ASR. The reason is that increasing the learning rate tends to make the model skip the local optimal value and get a more likely worse value.

Therefore, combining the results in these 2 subfigures, we choose the learning rate 0.001 as the default setting in § 5.2 because using 0.001 achieves the best BA and ASR. The epoch number is set to 5 because ASR is not decreased after 5 epochs and the BA is already good.

**Resetting fraction.** Resetting fraction measures the number of neurons that are reset by NONE. Specifically, NONE first sorts the probabilities that the neuron has activation values larger than 0 and then resets the neurons whose probabilities is in top $r_1\%$ in each layer. Using a smaller resetting fraction makes NONE to detect compromised neurons more conservatively (only labeling and resetting the most likely compromised neurons). To measure the effect of resetting fraction on defense performance of NONE, we obtain the BA and ASR of models at different resetting fractions from 0.5% to 15%. The results are shown in Fig. 9, where the legend shows different resetting fraction values.

From the results in Fig. 9(a), it is obvious that when we use a larger resetting fraction and reset more neurons, the final BA is lower. The reason is that after we reset neurons, some good features learned by the model are lost, which decreases the final BA. When we reset more neurons (i.e., using a larger resetting fraction), the model loses more high quality features and decreases more BA. Therefore, to avoid losing too much BA, the resetting fraction is recommended to be small.

**Table 13:** Results on Different Selection Thresholds.

| Number of Neurons | Single Target Attack | | Label Specific Attack | |
|---|---|---|---|---|
| | BA | ASR | BA | ASR |
| top 1 | 93.11% | 1.03% | 93.31% | 0.96% |
| top 0.5% | 93.10% | 1.07% | 93.32% | 0.96% |
| top 1% | 93.13% | 1.07% | 93.37% | 0.95% |
| top 10% | 93.11% | 1.06% | 93.29% | 1.04% |
| top 30% | 93.14% | 1.04% | 93.26% | 41.07% |
| top 50% | 93.08% | 1.04% | 93.18% | 60.12% |
| top 100% | 93.05% | 1.07% | 93.14% | 71.78% |

**Table 14:** Results on Different $\lambda_l$ and $\lambda_h$.

| $\lambda_h$ | BA | ASR | $\lambda_l$ | BA | ASR |
|---|---|---|---|---|---|
| 0.3 | 93.22% | 1.07% | 0.1 | 93.18% | 1.13% |
| 0.5 | 93.05% | 1.14% | 0.3 | 93.08% | 0.99% |
| 0.7 | 93.13% | 1.11% | 0.5 | 93.18% | 1.11% |
| 0.9 | 93.12% | 1.11% | 0.7 | 93.11% | 1.03% |

Furthermore, in Fig. 9(b), we find that different resetting fractions do not affect the ASR of models after a certain threshold (i.e., 3%). Because when the resetting fraction is large, NONE can successfully detect almost all compromised neurons. Increasing the resetting fraction does not help NONE to detect more compromised neurons.

Based on the above conclusions, we set the default resetting fraction as 3% because using resetting fraction 3% requires changing fewer neurons, achieving high BA and low ASR.

**Selection threshold.** When detecting poisoning samples, we only use the neurons whose compromised values are larger than the values of a portion of neurons in the same layer and we call this portion as selection threshold. To fully understand the impact of this threshold, we vary the threshold from 1 neuron to 100% neurons in the dataset and collect the corresponding BA and ASR under different attack settings. We test the single target BadNets attack and the label specific BadNets attack. We then show the results in Table 13, where the first column shows the threshold and the following columns show the results against the BadNets.

As the results show, when we increase the selection threshold, the ASR of the label specific BadNets attack significantly increases when the threshold is larger than 10%. This is because only a few neurons in the model are compromised. If the selection threshold is larger than the number of compromised neurons, NONE chooses many benign neurons to identify whether a sample is malicious or not, which introduces more noise and reduces the detection accuracy because benign neurons are not sensitive to Trojan behavior. Furthermore, the label specific BadNets attack specifies many different labels as target labels, making the attack stealthy and detecting the attack more difficult. Therefore, with the increase of the selection threshold, the defense performance becomes worse.

However, we observe that the ASR of the single target standard Trojan attack is not correlated with the selection threshold, showing the robustness of NONE to selection threshold against the single target Trojan attack. This is due to the fact that the single target BadNets attack only focuses on one label, making the malicious behavior more obvious, thus reducing the impact of introduced noise and still achieving a low ASR.

For the BA, we find that the BA against both the single target BadNets attack and the label specific BadNets attack is stable. Although using a lower selection threshold may allow NONE to filter out malicious samples conservatively (i.e., only use the most likely compromised neurons to detect malicious samples), enabling NONE to train the model on most of the data and achieve good BA results. Choosing a higher selection threshold does not decrease the BA significantly. Because considering there are a large number of benign samples in the dataset, even a higher selection threshold introduces more benign neurons (i.e., noise) to identify malicious samples and reduces the number of benign samples for finetuning, NONE still has enough benign samples for training and achieves similar BA results as using low selection thresholds.

Therefore, considering both BA and ASR, we set the selection threshold as 10% to avoid the ASR increasing significantly.

**Parameters in compromised neurons identification.** As mentioned in § 4 of the main paper, we use an alternative implementation to evaluate our design. We first obtain two clusters of samples according to their final layer probability outputs (the value in the probability vector). Subsequently, we classify the samples whose probability values are lower than a threshold $\lambda_l$ to the first cluster (i.e., low confidence samples) and classify the samples whose probability values are higher than $\lambda_h$ to the second cluster (i.e., low confidence samples). Then, we use the gap between two clusters to measure the linearity of each neuron. If a neuron has high linearity (i.e., top $r_1$ in a layer), then we

**Table 15:** Comparisons on Efficiency.

| Method | Runtime | Overhead |
|---|---|---|
| Native training | 2898.4s | N/A |
| AC | 4459.7s | 53.86% |
| ABL | 3197.4s | 10.31% |
| NONE | 3149.7s | 8.60% |

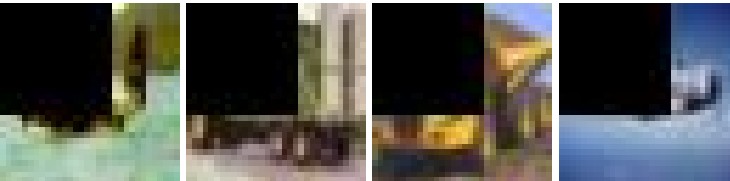

**Fig. 10:** Trojan Inputs with Large Triggers.

consider it as a compromised neuron. In this process, $\lambda_h$ and $\lambda_l$ determine the selection of the high confidence

samples and the low confidence samples that affect the defense performance of NONE. Therefore, to fully understand the impacts of them, We vary $\lambda_h$ and $\lambda_l$ values, and obtain the corresponding ASR and BA. By default, we use $\lambda_h$ as 0.9 and $\lambda_l$ to be 0.1 when the other parameter is changing.

Table 14 shows the results with different $\lambda_h$ and $\lambda_l$ settings. The results indicate that there is no obvious correlation between the performance of NONE and parameters (i.e., $\lambda_l$ and $\lambda_h$). As the results show, the ASR of models is always around 1.11% under different parameter settings. And the difference between the highest BA and the lowest BA is 0.17% which is quite small. Therefore, NONE is not sensitive to $\lambda_l$ and $\lambda_h$, which improves the usability of NONE.

### 8.9 Efficiency

We compare the total training time of native training, AC [22], ABL [9], and our method on the CIFAR-10 dataset with ResNet18. The results are shown in Table 15. The epoch number (i.e., 100) and batch size (i.e., 128) for different methods are the same. The ASR and BA are consistent with results in Table 1. We run each method with five trails and report the average time. All methods are run on the same device specified in § 5 of the main paper. Thus, our method is efficient.

### 8.10 Adaptive Attack

In this paper, we assume that attackers can poison the training data but have no control over the training procedure, e.g., the training algorithm, code, and hardware. This is consistent with existing work [24, 22, 9, 58]. It is hard for attackers to conduct adaptive attacks under the threat model because they can not directly control the training of the model, instead, NONE will be in charge of the training process. Therefore, we relax the threat model and consider the adaptive attacker in a code-poisoning attack [57], which requires extra capability from the adversary, i.e., modifying the training procedure.

In the considered adaptive code-poisoning attacks, the adversary goal is to train a Trojaned model with low linearity and try to evade the defense of NONE. However, under our threat model, the adversary can only poison the data but cannot modify the training process of NONE, which makes reducing the model's internal linearity almost impossible. Therefore, we relax the threat model for attackers and allow the attacker to control the training process of the model. We also assume the defender can access both the training data and the trained model. The defender tries to use NONE to eliminate Trojans injected in the model trained by the attacker.

Then, we design an adaptive loss that minimizes the activation difference between benign samples and corresponding Trojan samples to achieve attack goals. The adaptive loss is defined in Eq. 9, where $x$ is benign sample and $\tilde{x}$ is the corresponding Trojan sample (i.e., the sample obtained by pasting trigger on $x$).

$$\mathcal{L}\left(F_\theta(x), y\right) + \mathcal{L}\left(F_\theta(\tilde{x}), y_t\right) + \alpha \sum \left(I_i(x) - I_i(\tilde{x})\right)^2 \tag{9}$$

**Table 16:** Adaptive Attack.

| $\alpha$ | Undefended | | NONE | |
|---|---|---|---|---|
| | BA | ASR | BA | ASR |
| 1e-4 | 90.06% | 100.00% | 88.48% | 67.89% |
| 1e-3 | 89.53% | 99.97% | 87.92% | 76.78% |
| 1e-2 | 89.03% | 99.91% | 86.50% | 86.20% |
| 1e-1 | 88.72% | 99.98% | 85.71% | 94.92% |

$y$ and $y_t$ are the label of benign sample $x$ and target label respectively. $F_\theta$ donates the final prediction of the model. $\mathcal{L}$ means the Cross-Entropy criterion. Meanwhile, $I_i$ is the feature on the i-th layer, and $\alpha$ is the weight that controls the influence of the third loss item. By design, the loss function minimizes the distance between activation values of benign samples and the corresponding Trojan samples, making the Trojan decision region more curve and complex. Trojan models trained with the adaptive loss should have low linearity and may evade the detection of NONE.

To measure whether the adaptive attack works, we first train a benign model and then fine-tune that model using adaptive loss when attackers use poisoned data to attack the model. The Trojan trigger we use in the attack is the watermarking trigger and the model is VGG16. The results are shown in Table 16. The results show that NONE does not always achieve good defense against adaptive attacks. For example, when $\alpha = 1e - 1$, the BA and the ASR of NONE is 85.71% and 94.92%, respectively. However, the BA and ASR of the model trained with NONE are lower than that of the undefended model, showing that NONE helps in training a better model.

## 8.11 Comparison with DBD

Besides existing defenses compared in § 5.2 (i.e., DP-SGD [24], NAD [38], AC [22], ABL [9]), we also compare NONE with another training time defense DBD [58]. DBD defends backdoor attacks by decoupling the end-to-end training process into three stages, i.e., self-supervised learning for the backdoor, supervised training for the fully-connected layers, and semi-supervised fine-tuning of the whole model. We use six different attacks (i.e., BadNets [1], Label-consistent [11], Blend [10], SIG [69], Filter [8], WaNet [68]) and the CIFAR-10 dataset. We report the BA and ASR of the native training, DBD, and NONE in Table 17. The average runtime of DBD and NONE are 18,988.4s and 3,149.7s, respectively. For all attacks, our method achieves higher BA than DBD. In addition, in five of six attacks, the ASR of NONE is lower than that of DBD. The results show that our method is more effective and efficient than DBD.

**Table 17:** Comparison to DBD [58].

| Attack | Undefended | | DBD | | NONE | |
|---|---|---|---|---|---|---|
| | BA | ASR | BA | ASR | BA | ASR |
| BadNets | 94.10% | 100.00% | 91.24% | 1.25% | **93.62%** | **1.07%** |
| Label-consistent | 94.73% | 83.42% | 91.08% | **1.87%** | **94.01%** | 2.14% |
| Blend | 94.62% | 99.86% | 92.03% | 1.96% | **94.21%** | **0.93%** |
| SIG | 94.34% | 99.08% | 91.55% | 1.51% | **93.79%** | **1.08%** |
| Filter | 91.08% | 99.34% | 88.75% | 1.42% | **89.87%** | **1.20%** |
| WaNet | 94.39% | 96.71% | 90.98% | 0.95% | **92.24%** | **0.69%** |

## 8.12 Comparison to More Defenses on Natural Trojan

Besides the results of comparison to DP-SGD on natural Trojan (§ 5.2), we compare NONE with more training-time defenses (i.e., DP-SGD [24], NAD [38], AC [22], ABL [9], DBD [58]) on natural Trojan [8]. The dataset used here is CIFAR-10, and DNNs are NiN and VGG16. As shown in Table 18, the average ASR of NONE is 33.07%, 2.41 times lower than the undefended model. However, the average ASR of DP-SGD, NAD, AC, ABL, and DBD are 75.4%, 80.43%, 77.45%, 79.32%, 77.99%, respectively. The results demonstrate all existing methods have high ASR when facing natural backdoors, while our method can reduce the ASR significantly.

**Table 18:** Comparisons to More Defenses on Natural Trojan.

| Network | Undefended | | DP-SGD | | NAD | | AC | | ABL | | DBD | | NONE | |
|---------|------------|------|--------|------|-------|------|-------|------|-------|------|-------|------|-------|------|
| | BA | ASR | BA | ASR | BA | ASR | BA | ASR | BA | ASR | BA | ASR | BA | ASR |
| NiN | 91.02% | 87.62% | 39.19% | 87.22% | 80.75% | 88.10% | 83.85% | 88.28% | 86.28% | 86.33% | 86.27% | 87.54% | **86.94%** | **34.21%** |
| VGG16 | 90.78% | 71.88% | 53.40% | 63.58% | 85.20% | 72.76% | 85.69% | 66.67% | **86.46%** | 72.32% | 86.38% | 68.45% | 81.83% | **31.49%** |

## 8.13 Generalization to Larger Models

To study NONE's generalization to larger models, we report its BA and ASR on ResNet34 [53] and Wide-ResNet-16 (WRN16) [70]. The results of two baseline methods (i.e., NAD [38] and ABL [9]) are also reported. The dataset used is CIFAR-10. The runtime overhead of NONE on ResNet34 and WRN16 are 10.15% and 9.73%, respectively. For both two models, NONE achieves higher BA and lower ASR than NAD and ABL. For ResNet34, the BA of NONE is 2.47% and 2.84% higher than NAD and ABL. The ASR of NONE for ResNet34 is also 1.45% and 0.19% lower than that of NAD and ABL. The results show that our method is scalable to larger models.

## 8.14 Generalization to Larger Datasets

To evaluate the generalization of NONE to larger datasets, we report the performance (i.e., BA, ASR, and Runtime) of native training and NONE on a ImageNet subset (200 classes with 100k images for training and 10k images for testing) from Li et al. [71]. The results can be found in Table 20. NONE achieves low ASR (i.e., 1.98%, 50.32 times lower than Native Training) with a high BA (i.e., 1.66% lower than native training). In addition, the overheads compared with native training is 13.86%.

**Table 19:** Generalization to Larger Models.

| Networks | NAD | | ABL | | Ours | |
|----------|-------|------|-------|------|----------|----------|
| | BA | ASR | BA | ASR | BA | ASR |
| ResNet34 | 90.54% | 2.67% | 90.17% | 1.41% | **93.01%** | **1.22%** |
| WRN16 | 86.73% | 5.96% | 84.70% | 5.04% | **88.28%** | **3.88%** |

**Table 20:** Generalization to Larger Datasets.

| Method | BA | ASR | Runtime |
|--------|------|------|---------|
| Native Training | 85.12% | 99.65% | 23.8h |
| NONE | 83.46% | 1.98% | 27.1h |