# OpenReview forum: "Training with More Confidence: Mitigating Injected and Natural Backdoors During Training"
_NeurIPS.cc/2022/Conference — NeurIPS 2022 Accept_

### Official Review · Reviewer_bmQS · 2022-07-04

**Rating:** 6
**Confidence:** 4
**Soundness:** 3 good
**Presentation:** 2 fair
**Contribution:** 3 good

**Summary:**

This paper proves that a backdoored DNN learns a hyperplane as the decision region, under some assumptions. The authors argue that the backdoored DNNs will increase linearity by introducing a large percentage of neurons activating on one piece of the activation function. Based on these understandings, the authors propose a new training-stage defense (i.e., NONE) that detects and removes potential poisoned samples and repairs compromised neurons. The proposed defense is evaluated on MNIST, GTSRB, CIFAR-10, ImageNet-10, and TrojAI datasets.

**Questions:**

Cons
1. Some statements need more details and justifications.
- 'An adaptive attack with slow poisoning can bypass such defenses' (Line 5-6, Page 1): What the ‘slow poisoning’ means? It seems that there is no support (references or experiments) for it.
- ‘These methods fail to defend against the natural Trojan. (Line 38-39, Page 2)’: No experiments about whether ABL cannot remove natural Trojan (I only see the results of DPSGD).
2. Missing assumptions in Theorem 3.3. I think Theorem 3.3 holds if and only if the Trojan is complete and accurate.
3. Please provide more details about two statements in the appendix (Line 591-593, Page 14).
4. Missing an important related work and baseline. It seems that [1] is the SOTA training-stage backdoor defense, which is not mentioned and discussed in this paper.
5. The organization of Section 4 is very poor. There is a very long paragraph in this section, which significantly increases the reading difficulty. Besides, many important details are missing. For example, how to do the separation (Line 13, Algorithm 1)? What are the technical details of Fisher’s linear discriminant analysis and Jenks natural breaks optimization algorithm?
6. It seems that hidden Trojan backdoor attack is a training-stage attack, which is not effective if training from the scratch (as suggested in [2]). How did the authors evaluate defenses under this attack?
7. Why did the authors compare their defense only with DPSGD under natural Trojans? I understand that DPSGD seems to be effective in defending against natural Trojans. However, this is not the reason for not comparing with other defenses.
8. I think it is necessary to go deeper into the internal mechanism of the proposed method. Specifically, according to Algorithm 1, the proposed method can detect poisoned or biased samples. Please provide the precision and recall of this stage so that the readers can understand how well this stage is.

Minor Comments
1. One typo in Line 91 (Page 2): ‘We’ should be ‘we’.
2. The ‘R’ should be ‘R^l’ in Line 143 (Page 4)?
3. Please recheck and cite the official version of all references (e.g., [3-6]).


References
1. Backdoor Defense via Decoupling the Training Process. ICLR, 2022.
2. Sleeper Agent: Scalable Hidden Trigger Backdoors for Neural Networks Trained from Scratch. arXiv, 2021.
3. Poisoning and backdooring contrastive learning. ICLR, 2022.
4. BadNL: Backdoor Attacks against NLP Models with Semantic-preserving Improvements. ACSAC, 2021.
5. BadNets: Identifying Vulnerabilities in the Machine Learning Model Supply Chain. IEEE Access, 2019.
6. BACKDOORL: Backdoor Attack against Competitive Reinforcement Learning. IJCAI, 2021.

**Limitations:**

Limited but sufficient.

**Strengths And Weaknesses:**

Pros
1. The topic is of sufficient significance and interest to NeurIPS audiences.
2. The codes are provided, which should be encouraged.
3. The authors try to provide theoretical support for their method, which should be encouraged.
4. The authors discuss the resistance to potential adaptive attacks, which should be encouraged.
5. In general, the idea is easy to follow, although many important details are missing.



Cons
1. Some statements need more details and justifications.
- 'An adaptive attack with slow poisoning can bypass such defenses' (Line 5-6, Page 1): What the ‘slow poisoning’ means? It seems that there is no support (references or experiments) for it.
- ‘These methods fail to defend against the natural Trojan. (Line 38-39, Page 2)’: No experiments about whether ABL cannot remove natural Trojan (I only see the results of DPSGD).
2. Missing assumptions in Theorem 3.3. I think Theorem 3.3 holds if and only if the Trojan is complete and accurate.
3. Please provide more details about two statements in the appendix (Line 591-593, Page 14).
4. Missing an important related work and baseline. It seems that [1] is the SOTA training-stage backdoor defense, which is not mentioned and discussed in this paper.
5. The organization of Section 4 is very poor. There is a very long paragraph in this section, which significantly increases the reading difficulty. Besides, many important details are missing. For example, how to do the separation (Line 13, Algorithm 1)? What are the technical details of Fisher’s linear discriminant analysis and Jenks natural breaks optimization algorithm?
6. It seems that hidden Trojan backdoor attack is a training-stage attack, which is not effective if training from the scratch (as suggested in [2]). How did the authors evaluate defenses under this attack?
7. Why did the authors compare their defense only with DPSGD under natural Trojans? I understand that DPSGD seems to be effective in defending against natural Trojans. However, this is not the reason for not comparing with other defenses.
8. I think it is necessary to go deeper into the internal mechanism of the proposed method. Specifically, according to Algorithm 1, the proposed method can detect poisoned or biased samples. Please provide the precision and recall of this stage so that the readers can understand how well this stage is.

I will increase my scores if the authors can (partly) address my concerns.

---

> ### Author Response · Authors · 2022-08-02
> **Response to Reviewer bmQS - Part 1**
>
> Thank you for your time and insightful comments. We have run all the suggested
> experiments. We hope the following new clarifications and results can address
> your concerns. We are willing to perform more experiments if you have further
> suggestions.
>
> **Q1:** What the ‘slow poisoning’ means? It seems that there is no support
> (references or experiments) for it.
>
> **A1:** Thanks for your insightful question. The ‘slow poisoning’ is a
> backdoor attack strategy that uses a small poisoning ratio so that the
> poisoning effect is not obvious. This strategy is used to cover the existence
> of poisoning samples, which can bypass some existing defenses (e.g., ABL). The
> experiments on the label-specific attack using this strategy can be found in
> Table 1. We will clarify.
>
> **Q2:** No experiments about whether ABL cannot remove natural Trojan.
>
> **A2:** Thanks for your valuable comment. The results of ABL for natural
> Trojan are reported in the following table. The dataset used here is CIFAR-10.
> The results confirm that ABL is not able to remove natural Trojan. We will add
> more results to the revised paper.
>
> Network | | Undefended | ABL
> ---- | ---|--- | ---
> NiN | BA/ASR(%) | 91.02/87.62 | 86.28/86.33
> VGG16 |BA/ASR(%) | 90.78/71.88 | 86.46/72.32
>
> **Q3:** Missing assumptions in Theorem 3.3.
>
> **A3:** Thanks for your helpful comment. We assume that the backdoor trigger
> can be written as $ \tilde{x} = (1-m) {\odot} x + m {\odot} t $ ($m$ and $t$
> are respectively the trigger mask and the trigger pattern), and the attack is
> successful. We will revise it in the next version.
>
> **Q4:** Please provide more details about two statements in the appendix (Line
> 591-593, Page 14).
>
> **A4:** Thanks for your suggestion. We will revise accordingly in the
> revision. For any sample $\tilde{x}$ in the hyperplane $\{Ax -b = 0\}$, it can
> be obtained by projecting another normal sample (i.e., samples that are not in the
> hyperplane) into the hyperplane. In fact, any sample $x$ satisfy $(E-A) x =
> (E-A) \tilde{x}$ can be transformed to $\tilde{x}$ via the projection
> specified by $m$ and $t$.
>
> **Q5:** Comparison to Huang et al. is missing.
>
> **A5:** Thanks for your constructive comment. The results of the comparison
> with Huang et al. can be found in the following table. The average running
> time of Huang et al. is 18988.4s, while that of NONE is 3149.7s. The dataset
> is CIFAR-10 and the model used is ResNet18. The results demonstrate NONE is
> more effective and more efficient than Huang et al. We will add the results to
> the revised version.
>
> Attack | | Undefended | Huang et al. | NONE
> ---- | ---|--- | --- | ---|
> BadNets | BA/ASR(%) | 94.10/100.00 | 91.24/1.25| 93.62/1.07 |
> Label-consistent |BA/ASR(%) | 94.73/83.42 | 91.08/1.87 | 94.01/2.14
> Blend |BA/ASR(%) |  94.62/99.86 | 92.03/1.96 | 94.21/0.93
> SIG |BA/ASR(%) | 94.34/99.08 | 91.55/1.51 | 93.79/1.08
> Filter |BA/ASR(%) |  91.08/99.34 | 88.75/1.42 | 89.87/1.20
> WaNet |BA/ASR(%) |  94.39/96.71 | 90.98/0.95 | 92.24/0.69
>
> Huang et al., Backdoor Defense via Decoupling the Training Process. ICLR 2022.
>
> **Q6:** The organization of Section 4 is very poor. There is a very long
> paragraph in this section, which significantly increases the reading
> difficulty.
>
> **A6:** Thank you for your helpful suggestion. We will split the long
> paragraph into short paragraphs.
>
> **Q7:** Many important details are missing. For example, how to do the
> separation (Line 13, Algorithm 1)?
>
> **A7:** Thanks for your helpful question. In line 13 of Algorithm 1, we do the
> separation via Fisher’s linear discriminant analysis. Will clarify in the
> revision.
>
> **Q8:** What are the technical details of Fisher’s linear discriminant
> analysis and Jenks natural breaks optimization algorithm?
>
> **A8:** Thanks for your valuable question.
>
> We separate the activation values into two clusters via Fisher’s linear
> discriminant analysis. In detail, we minimize the variance within clusters
> $\sigma_{within}$ and maximize the variance between clusters
> $\sigma_{between}$. The process is implemented by Jenks natural breaks
> optimization, which is an iterative optimization method that finds the
> minima/maxima of $\sigma_{within}/\sigma_{between}$. We will add more details.

---

> > ### Author Response · Authors · 2022-08-02
> > **Response to Reviewer bmQS - Part 2**
> >
> > **Q9:** It seems that hidden Trojan backdoor attack is a training-stage
> > attack, which is not effective if training from the scratch (as suggested in
> > [2]). How did the authors evaluate defenses under this attack?
> >
> > **A9:** Thanks for your insightful question.
> >
> > Most existing training defenses [8,21,22,33] assume the attacker can only
> > poison the data before training the model. In this paper, we also assume the
> > attacker poisons the training data before the training. Hidden Trojan backdoor
> > attack shares the same threat model and assumes the poisoned dataset is
> > generated before the training. As pointed out in the Hidden Trojan backdoor
> > attack [43] and Sleeper Agent [Souri et al.], the hidden Trojan backdoor attack
> > is not effective in training from scratch. It works in the transfer learning
> > scenario (i.e., fine-tuning pre-trained models on poisoned data). For the
> > experiments of hidden Trojan backdoor attacks, we fine-tune the pre-trained
> > models on poisoned training data and use NONE to defend it.
> >
> > In Section 8.10 (Appendix), we also reported the results of the training-stage
> > poisoning attack (i.e., the attackers continuously providing poisoning data
> > during the training process) DBD (Xie et al.) on the federated learning
> > scenario. The results are summarized in the following table.
> >
> > Dataset | | Undefended | NONE
> > ---- | ---|--- | ---
> > MNIST | BA/ASR(%) | 99.22/99.15 | 98.60/0.13
> > CIFAR-10 |BA/ASR(%) | 80.31/43.23 | 78.67/4.44
> >
> > More details can be found in Section 8.10. The results show that NONE can
> > generalize to defend training stage attack.
> >
> > Souri et al., Sleeper Agent: Scalable Hidden Trigger Backdoors for Neural Networks Trained from Scratch. arXiv 2021.
> >
> > Xie et al., DBA: Distributed Backdoor Attacks against Federated Learning. ICLR 2020.
> >
> > **Q10:** Why did the authors compare their defense only with DPSGD under
> > natural Trojans? I understand that DPSGD seems to be effective in defending
> > against natural Trojans. However, this is not the reason for not comparing
> > with other defenses.
> >
> > **A10:** The following table provides comparisons to more existing defenses.
> > The dataset used here is CIFAR-10. The results demonstrate all existing
> > methods have high ASR when facing natural backdoors, while our method can
> > reduce the ASR significantly.
> >
> > Arch | | Undefended | DPSGD | NAD| AC| ABL| Decoupling| NONE
> > ---- | ---|--- | --- | ---|---|---|---|---|
> > NiN | BA/ASR |  91.02/87.62 | 39.19/87.22 | 80.75/88.10 |83.85/88.28 | 86.28/86.33 |86.27/87.54 |86.94/34.21 |
> > VGG16 |BA/ASR | 90.78/71.88 | 53.40/63.58 | 85.20/72.76| 85.69/66.67 |86.46/72.32 |86.38/68.45 |81.83/31.49 |
> >
> > **Q11:** I think it is necessary to go deeper into the internal mechanism of
> > the proposed method. Specifically, according to Algorithm 1, the proposed
> > method can detect poisoned or biased samples. Please provide the precision and
> > recall of this stage so that the readers can understand how well this stage
> > is.
> >
> > **A11:** Thanks for your helpful comment. The detailed precisions and recalls
> > are shown in the following table. The dataset used here is CIFAR-10.
> >
> > Attack |  | NiN | VGG16 | ResNet18
> > ---- | --- | ---| ---| ---
> > BadNets  | Precision/Recall (%) | 99.60/99.64 | 99.96/100.00 | 99.84/99.92
> > Label-consistent | Precision/Recall (%) | 98.80/99.20 | 100.00/100.00 | 100.00/100.00
> >
> > We will add more results in the revised version.
> >
> > **Q12:** Other minor comments.
> >
> > **A12:** Thank you for your helpful comment. We will modify it accordingly.

---

> > > ### Comment · Reviewer_bmQS · 2022-08-03
> > > **Post-rebuttal Comments**
> > >
> > > Thank you for adequtely addressed my concerns. As such, I increase my score to 'weak accept'.

---

> > > > ### Author Response · Authors · 2022-08-03
> > > > **Thanks**
> > > >
> > > > Thank you very much for your support and feedback. We will make sure the new results and clarifications are properly incorporated into the revised version.

---

### Official Review · Reviewer_75rW · 2022-07-10

**Rating:** 5
**Confidence:** 4
**Soundness:** 2 fair
**Presentation:** 3 good
**Contribution:** 2 fair

**Summary:**

This paper demonstrated the backdoor-related neurons in the DNN models form a hyperplane surface as the decision region to the backdoor target label. Based on this understanding, the authors proposed a robust training algorithm,  NONE (NON-LinEarity), that identifies linear decision regions, filters out inputs that are potentially poisoned, and resets affected neurons to enforce non-linear decision regions during training.


**Questions:**

I have a few concerns regarding the proposed method and experiments：

- The finding that shortcuts or linearly-separable characteristics for backdoor attacks are relatively well-known. The same observation has also been done in works (https://arxiv.org/abs/2106.07214) and (https://arxiv.org/pdf/2111.00898.pdf) for backdoor attacks, so it may be worth clarifying that this finding is not completely novel.

- The authors claim that the hyperplane surface formed in backdoored model is because of the piece-wise linear function (i.e. Relu) and there exists one and only one hyperplane in the input space that corresponds to all poisoning samples. What evidence could support your argument and why the reason there is only exist one hyperplane? Whether such linear separability can be induced by the kind of model architecture, input dimensionality, and other activation functions?

- The authors have shown that the traditional backdoor attack methods such as BadNets attack, produce linear separability against a deep neural network. Nevertheless, It may be worth investigating the possibility of crafting “adaptive” attacks [1,2,3] that confuse both clean and backdoor features together when considering separating hyperplanes. In other words, whether the hyperplane surface assumption still held under such cases?

- For the backdoor remove training, why reset the neuron during training? Instead of resetting after training?

- The authors should add a comparison of their computational complexity.


[1] Shafahi A, Huang W R, Najibi M, et al. Poison frogs! targeted clean-label poisoning attacks on neural networks. NeurIPS, 2018.
[2] Nguyen T A, Tran A. Input-aware dynamic backdoor attack. NeurIPS 2021.
[3] Doan K, et al. Backdoor attack with imperceptible input and latent modification. NeurIPS 2021.


**Limitations:**

The linearly-separable assumption as an underlying explanation for the effectiveness of backdoor attacks is interesting. However, this work in its current form still lacks sufficient analysis to support this conclusion.

**Strengths And Weaknesses:**

The paper is well written and easy to understand.

---

> ### Author Response · Authors · 2022-08-02
> **Response to Reviewer 75rW - Part 1**
>
> Thank you for your time and valuable comments. We hope the following
> clarifications can address your concerns.
>
> **Q1:** The finding that shortcuts or linearly-separable characteristics for
> backdoor attacks are relatively well-known.
>
> **A1:** Thank you for the suggested papers. We respectively disagree.
>
> $\bullet$ Cina et al. use the loss curve in the training process to study
> **the process of backdoor learning** under the lens of incremental learning
> and influence functions. It tries to find the key factors that enable backdoor
> learning, e.g., model size, poisoning ratio, and trigger size and visibility.
> Our theoretical analysis is training agnostic. It models the data distribution
> of benign and poisoning samples in the input space. We were not able to
> identify similar conclusions in the mentioned paper. Please let us know if a
> more detailed comparison is needed.
>
> $\bullet$ Yu et al. focus on availability attacks, in which the perturbations
> are imperceptible. In the backdoor attack setting, triggers can be large
> patches and the pattern can be complex. We agree that these two types of
> attacks share some common attacks. However, they are fundamentally different.
> Contradictory to the linear separable characteristics of availability attacks,
> researchers have shown that many backdoor attacks are not linearly separable
> [49]. While our finding is that the backdoor region is a hyperplane, their
> finding is the perturbations of availability attacks are linear separable,
> which are not the same. Moreover, backdoor triggers are NOT shortcuts -- which
> is defined as: "Shortcuts are spurious features that are correlated with
> target labels but do not generalize on test data". However, backdoor attacks,
> e.g., feature-space attacks [Cheng et al.] and natural Trojans [36] have a
> high attack success rate on test data.
>
> Cheng et al., Deep Feature Space Trojan Attack of Neural Networks by
> Controlled Detoxification. AAAI 2021.
>
> $\bullet$ In addition, existing work is based on empirical observations. We
> provide a theoretical analysis and proof of our findings. The theory models
> the poisoning data distribution and works on different model architectures and
> attacks.
>
> **Q2:** The authors claim that the hyperplane surface formed in backdoored
> model is because of the piece-wise linear function (i.e. ReLU) and there
> exists one and only one hyperplane in the input space that corresponds to all
> poisoning samples. What evidence could support your argument and why the
> reason there is only exist one hyperplane?
>
> **A2:** We would like to clarify that the hyperplane is not **because of** the
> piece-wise linear function. The hyperplane exists because of the backdoor
> attacks that poison the dataset. It is general to all different models using
> different activation functions. We provide theoretical proof, showing why
> there exists one and only one such hyperplane in Section 8.1 of the appendix.
> Our analysis of piece-wise linear function is to show **how** popular model
> architectures learn such hyperplanes in practice. In Figure 5 and Figure 6
> (Appendix, Supplementary material), we also show empirical evidence to support
> our analysis, using different model architectures and layers, in addition to
> our theoretical analysis and proof. We will revise it to clarify.
>
>
> **Q3:** Whether such linear separability can be induced by the kind of model
> architecture, input dimensionality, and other activation functions?
>
> **A3:** Our analysis and proof (Section 8.1) do not depend on model
> architectures and activation functions. Our empirical results also show that
> NONE is robust to different model architecture, input dimensionality, and
> activation functions.
>
> $\bullet$ We have reported the results under different activation functions
> (LeakyReLU, ELU, Tanhshrink, Softplus) in Table 8 in the Appendix.
>
> $\bullet$ We also already demonstrated the results under different model
> architectures (NiN, VGG11, VGG16, and ResNet18, see Table 1-4 in the main paper
> and Fig.5-6 in the appendix).
>
> $\bullet$ Please see Table 9 in the Appendix for the results under different
> input dimensionality (from 1x28x28 to 3x224x224).
>
> The results show the generalization of NONE in different settings (i.e., model
> architectures, input dimensionalities, and activation functions).
>
> **Q4:** Whether the hyperplane surface assumption still held under “adaptive”
> attacks?
>
> **A4:** Thanks for your valuable question. We would like to clarify that the
> hyperplane analysis is not an assumption but a property that can be proved.
> Please see Section 8.1 (Appendix).

---

> > ### Author Response · Authors · 2022-08-02
> > **Response to Reviewer 75rW - Part 2**
> >
> > **Q5:** For the backdoor remove training, why reset the neuron during training?
> > Instead of resetting after training?
> >
> > **A5:** Thanks for your helpful question. Directly resetting backdoor-related
> > neurons after training will hurt the benign accuracy of model [35].
> > Therefore, we reset the neurons during training so that the benign accuracy of
> > the models can be maintained via optimizing on clean samples.
> >
> > **Q6:** The authors should add a comparison of their computational complexity.
> >
> > **A6:** Thanks for your constructive comment. We compare the total training
> > time of AC, ABL, and our method on the CIFAR-10 dataset with the ResNet18
> > model. The results are shown in the following table:
> >
> > Method | Runtime |
> > ---- | --- |
> > Native Training | 2898.4s |
> > AC | 4459.7s |
> > ABL | 3197.4s |
> > Ours | 3149.7s |
> >
> > The epoch number (i.e., 100) and batch size (i.e., 128) for different methods
> > are the same. The ASR and BA are consistent with the reported results in Table 1. Please kindly see more details on the comparison of runtime in Section 8.9
> > of the Appendix. More results can be found in answers to Review obsx Q3.

---

> > > ### Comment · Reviewer_75rW · 2022-08-08
> > > **Response to rebuttal**
> > >
> > > Thanks for the detailed response which partially addresses my concerns. As such, I have increased my score to 5.

---

> > > > ### Author Response · Authors · 2022-08-08
> > > > **Thanks**
> > > >
> > > > Thanks for your support and feedback. We will make sure the new results and clarifications are properly incorporated into the revised version.

---

### Official Review · Reviewer_obsx · 2022-07-11

**Rating:** 7
**Confidence:** 4
**Soundness:** 3 good
**Presentation:** 3 good
**Contribution:** 4 excellent

**Summary:**

The paper proposes a backdoor defense based on the observation that trojan and benign neurons exhibit different behavior in the victim model. Their defense assumes an attacker who can poison a subset of the training data and its labels. They design a training algorithm that detects and suppresses neurons that form shortcuts between classes, which they describe as "linear decision boundaries". Empirical results are provided for image classification datasets, showing that their training algorithm NONE has considerable improvements over existing defenses against three poisoning-based backdoor attacks. Finally, the authors show an ablation study over different trigger sizes and poisoning rates which show that NONE is robust in most cases.

**Questions:**

What is the runtime of NONE? How does it scale to larger models or datasets such as ImageNet?

How many inputs are removed by NONE on datasets such as CIFAR-10? What is the precision and recall (with regards to malicious and clean inputs) of this removal process?

Have you looked into the performance of NONE against backdoor attacks that can poison the training code?

**Limitations:**

-

**Strengths And Weaknesses:**

** Strengths

* Novelty. Defenses against poisoning-based backdoor attacks are highly relevant. Detecting and suppressing malicious neurons during training is a novel idea that shows promising results.

* High effectiveness. It appears that NONE is a promising defense that clearly outperforms many existing defenses. The drop in clean accuracy is also often low, (<=3%) making it a practically relevant defense.

* Ablation studies. The defense was evaluated on multiple datasets against three attacks showing promising results everywhere. The authors also show ablation studies over different trigger sizes and poisoning rates.

* Open-source code.

** Weaknesses

Overall, the paper was an interesting read and I found only minor weaknesses in the paper's presentation. The results and intuition are convincing and I believe they are of interest to the research community.

* Formalization. Many symbols in Algorithm 1 are only defined informally in the text, but it would improve clarity if they were formalized (e.g., the separate, norm, and init function). A table explaining the symbols would help readability a lot.

* Minor. Please break a single, long paragraph into multiple smaller paragraphs to enhance readability. Some citations are missing (e.g., for all datasets such as TrojAI [1])

[1] Karra, Kiran, Chace Ashcraft, and Neil Fendley. "The trojai software framework: An opensource tool for embedding trojans into deep learning models." arXiv preprint arXiv:2003.07233 (2020).

---

> ### Author Response · Authors · 2022-08-02
> **Response to Reviewer obsx - Part 1**
>
> Thank you very much for your thoughtful comments and recognition of the
> novelty and significance of our work. We hope the following results and
> clarifications can address your concerns.
>
> **Q1:** Many symbols in Algorithm 1 are only defined informally in the text. A
> table explaining the symbols would help readability a lot.
>
> **A1:** Thanks for your suggestion! The following table summarizes all symbols,
> and we will add it in the revised version:
>
> Symbol | Meaning |
> ---- | --- |
>  $D$ | Training data |
>  $E$ | Maximal epoch |
>  $e$ | Current epoch |
>  $M$ | Model |
>  $n$ | Neuron |
>  $A$ | Activation values |
>  $A_n$ | Activation values on neuron $n$ |
>  $C$ | Compromised Neurons |
>  $B_n$ | The cluster of smaller values in $A_n$ |
>  $O_n$ | The cluster of larger values in $A_n$ |
>  $\mu$ | Mean value of $B_n$ |
>  $\sigma$ | Standard deviation value of $B_n$|
>  $i$ | Input sample |
>  $i_n$ | The activation value of input sample $i$ on neuron $n$ |
>
>
> **Q2:** Please break a single, long paragraph into multiple smaller paragraphs
> to enhance readability. Some citations are missing.
>
> **A2:** Thanks for your constructive comment! We will split the long paragraph
> in Section 4 into multiple smaller paragraphs in the revised version. We will
> add the missing citations for the datasets in the revised version.
>
> **Q3:** What is the runtime of NONE? How does it scale to larger models or
> datasets such as ImageNet?
>
> **A3:** Thanks for your valuable question. We will add more results (and
> comparison), including the following, in our revised version.
>
> $\bullet$ Comparison of runtime: We compare the total training time of native,
> AC, ABL, and our method on the CIFAR-10 dataset with ResNet18 using the same
> configuration, e.g., epoch number (i.e., 100) and batch size (i.e., 128). The
> results are shown in the following table:
>
> Method | Runtime | Overheaed |
> ---- | --- | --- |
> Native Training | 2898.4s | N/A |
> AC | 4459.7s | 53.86% |
> ABL | 3197.4s | 10.31% |
> Ours | 3149.7s | 8.60% |
>
>
> The ASR and BA are consistent with the results in Table 1. Please kindly see
> more details on the comparison of runtime in Section 8.8 of the Appendix
> (Supplementary material). In short, compared with native training, our
> overhead is 8.60%, which is 45.26% and 1.71% less than AC and ABL,
> respectively.
>
> $\bullet$ Scale to larger models: The results on larger models can be found
> below. The dataset and the attack used are CIFAR-10 and BadNets. The overheads
> compared with native training on ResNet34 and WRN-16-1 are 10.15% and 9.73%,
> respectively. The results show that our method is scalable to larger models.
>
> Network |  | NAD | ABL | Ours
> ---- | --- | ---| ---| ---
> ResNet34  | BA/ASR (%) | 90.54/2.67 | 90.17/1.41 | 93.01/1.22
> WRN-16-1 | BA/ASR (%) | 86.73/5.96 | 84.70/5.04 | 88.28/3.88
>
> $\bullet$ Scale to larger datasets: Our preliminary results on a ImageNet subset
> (200 classes with 100k images for training and 10k images for testing) from Li
> et al. (ICCV 2021) are shown in the following table:
>
> Method  | BA | ASR | Runtime
> ----  | ---| ---| ---
> Native Training  | 85.12% | 99.65% | 23.8h
> Ours | 83.46% | 1.98% | 27.1h
>
> The overheads compared with native training is 13.86%. Thus, our method is
> scalable to large datasets.
>
> Li et al., Invisible Backdoor Attack with Sample-Specific Triggers. ICCV 2021.
>
> **Q4:** How many inputs are removed by NONE on datasets such as CIFAR-10? What
> is the precision and recall (with regards to malicious and clean inputs) of
> this removal process?
>
> **A4:** The detailed precisions and recalls are shown in the following table.
> The dataset used here is CIFAR-10.
>
> Attack |  | NiN | VGG16 | ResNet18
> ---- | --- | ---| ---| ---
> BadNets  | Precision/Recall (%) | 99.60/99.64 | 99.96/100.00 | 99.84/99.92
> Label-consistent | Precision/Recall (%) | 98.80/99.20 | 100.00/100.00 | 100.00/100.00
>
> We will add more results to the revised version.

---

> > ### Author Response · Authors · 2022-08-02
> > **Response to Reviewer obsx - Part 2**
> >
> > **Q5:** Have you looked into the performance of NONE against backdoor attacks
> > that can poison the training code?
> >
> > **A5:** Thank you for the suggestion! In this paper, we focus on the
> > data-poisoning threat model and assume that attackers can only poison the
> > training data, but have no control over the training procedure including the
> > training algorithm, code, and hardware, consistent with existing work
> > [8,21,33]. Data-poisoning attack is a popular, practical, and stealthy threat
> > model. Because inspecting all training data is more challenging than
> > inspecting the training code -- especially for DNN, the training code is
> > relatively short (compared with programs like Firefox) while the size of the
> > training data is often huge.
> >
> > The threat model of code-poisoning attack, which requires extra capability
> > from the adversary -- modifying the training procedure, is different from
> > ours. We consider it an adaptive attack in Section 8.11 (Appendix,
> > Supplementary material). In detail, we relax the threat model for attackers
> > and allow the attacker to control the training process of the model (e.g.,
> > modifying training code). We also assume the defender can access both the
> > training data and the trained model. The defender tries to use NONE to
> > eliminate Trojans injected in the model trained by the attacker. Results show
> > that with more power, the adversary can bypass such a training-time defense
> > technique. We believe this is a general drawback of all training-time defense
> > mechanisms. We will discuss it in our revised version.

---

### Author Response · Authors · 2022-08-02
**Rebuttal Summary**

We sincerely thank all reviewers for their valuable comments and precious
time. We provide our responses below to address the concerns. Please let us
know if anything is still unclear. We are happy to answer more questions and
conduct more experiments if needed.

---

### Author Response · Authors · 2022-08-08
**Revision Summary**

We thank all reviewers again for their insightful questions and suggestions. Below is our revision summary:

**[Section 2]** We added clarification for "slow poisoning", following the suggestion of Reviewer obsx and Reviewer bmQS.

**[Section 3]** We added the assumptions in Theorem 3.3, following the suggestion of Reviewer obsx and Reviewer bmQS. We added the clarification about the relationship between the hyperplane and the piece-wise linear function, following the suggestion of Reviewer obsx and Reviewer 75rW.

**[Section 4]** We splitted the long paragraph in Section 4 into multiple smaller paragraphs, following the suggestion of Reviewer obsx and Reviewer bmQS.

**[Appendix]** We added the table to summarize all symbols , following the suggestion of Reviewer obsx.

**[Appendix Section 8.1]** We added more technical details about the separation, following the suggestion of Reviewer bmQS.

**[Appendix Section 8.2]** We revised the two statements in the appendix (Line 591-593, Page 14) to make them more clear, following the suggestion of Reviewer bmQS.

**[Appendix Section 8.11]** We added the discussion about performance against code poisoning attack, following the suggestion of Reviewer obsx.

**[Appendix Section 8.12]** We added results for the precision and recall of poisoned samples identification stage, following the suggestion of Reviewer obsx and Reviewer bmQS.

**[Appendix Section 8.13]** We added the results of comparison to Huang et al., following the suggestion of Reviewer bmQS.

**[Appendix Section 8.14]** We added the results of comparisons with more defenses on natural Trojan, following the suggestion of Reviewer bmQS.

**[Appendix Section 8.15]** We added the results on larger models, following the suggestion of Reviewer obsx.

**[Appendix Section 8.16]** We added the results on larger datasets, following the suggestion of Reviewer obsx.

**[Other minor issues]** We fixed minor issues, following the suggestion of Reviewer bmQS.

---

### Meta-Review · Area_Chair_vLeS · 2022-08-24

**Recommendation:** Accept
**Confidence:** Certain

**Metareview:**

This paper introduces a new algorithm to mitigate backdoors in neural network models.
The reviewers agreed this paper proposes an interesting defense that is well motivated,
carefully evaluated with many ablation studies, and highly effective. The weaknesses
raised have also been mitigated in the rebuttal period and the paper is generlly strong.

**Award:**

No

---

### Decision · Program_Chairs · 2022-09-14

Accept